# SHE: Streaming-media Hashing Retrieval

**Ruitao Pu** [1]  **Yang Qin** [1]  **Xiaomin Song** [2]  **Dezhong Peng** [1 2 3]  **Zhenwen Ren** [4]  **Yuan Sun** [1 5]

## Abstract

Recently, numerous cross-modal hashing (CMH) methods have been proposed, yielding remarkable progress. As a static learning paradigm, existing CMH methods often implicitly assume that all modalities are prepared before processing. However, in practical applications (such as multimodal medical diagnosis), it is very challenging to collect paired multi-modal data simultaneously. Specifically, they are collected chronologically, forming streaming-media data (SMA). To handle this, all previous CMH methods require retraining on data from all modalities, which inevitably limits the scalability and flexibility of the model. In this paper, we propose a novel CMH paradigm named Streaming-media Hashing rEtrieval (SHE) that enables parallel training of each modality. Specifically, we first propose a knowledge library mining module (KLM) that extracts a prototype knowledge library for each modality, thereby revealing the commonality distribution of the instances from each modality. Then, we propose a knowledge library transfer module (KLT) that updates and aligns the new knowledge by utilizing the historical knowledge library, ensuring semantic consistency. Finally, to enhance intra-class semantic relevance and inter-class semantic disparity, we develop a discriminative hashing learning module (DHL). Comprehensive experiments on four benchmark datasets demonstrate the superiority of our SHE compared to 14 competitors.

[1]School of Computer Science, Sichuan University, Chengdu, China [2]Sichuan National Innovation New Vision UHD Video Technology Co., Ltd., Chengdu, China [3]Tianfu Jincheng Laboratory, Chengdu, China [4]Southwest University of Science and Technology, Mianyang, China [5]National Key Laboratory of Fundamental Algorithms and Models for Engineering Numerical Simulation, Sichuan University, Chengdu, China. Correspondence to: Yuan Sun <sunyuan_work@163.com>.

*Proceedings of the 42nd International Conference on Machine Learning*, Vancouver, Canada. PMLR 267, 2025. Copyright 2025 by the author(s).

## 1. Introduction

Recently, with the rapid development of smart devices and social media, multimedia data (such as images, audio, video, etc) has shown an explosive growth trend. Cross-modal retrieval (CMR) for massive and complex multimedia data has attracted widespread attention in academia and industry (Zhu et al., 2023; Liang et al., 2024). CMR aims to use data from one modality as the query to retrieve semantically related samples in another modality. However, due to different feature distributions, there is an inherent heterogeneity gap between different modalities. How to establish effective cross-modal mapping relationships to achieve accurate semantic retrieval between different modalities is a key challenge.

To bridge the heterogeneity gap, numerous CMR methods (Sun et al., 2024; Li et al., 2025b; Liu et al., 2024)have been proposed to capture the shared semantic information between different modalities, which could be roughly divided into unsupervised (Hu et al., 2022; Li et al., 2025a; 2024) and supervised methods (Wang et al., 2024a;b; Pu et al., 2025b). Among them, unsupervised CMR methods aim to directly explore the semantic similarity relationships between multi-modal data according to the original features. However, due to the lack of available semantic labels, unsupervised methods suffer from a performance bottleneck. Thus, many supervised CMR methods are proposed that utilize label information to guide the learning of semantic similarity relationships between instances, thereby obtaining the discriminative common representations. Most of these methods employ a joint learning manner to learn common representations for multi-modal data. In other words, they necessitate that data from all modalities be available simultaneously and collaboratively incorporated to learn a common representation. Although these methods have demonstrated promising performance, their success heavily relies on the implicit assumption of complete data availability. In practical application scenarios, it is difficult to collect data from all modalities simultaneously, such as emergency medical aid. Due to the asynchronous characteristics of multi-modal data collection, these data from different modalities are often continuously collected and processed at different time points, thus forming streaming-media data (SMA). When dealing with SMA, the aforementioned methods must frequently retrain all the sub-networks for all modalities, which

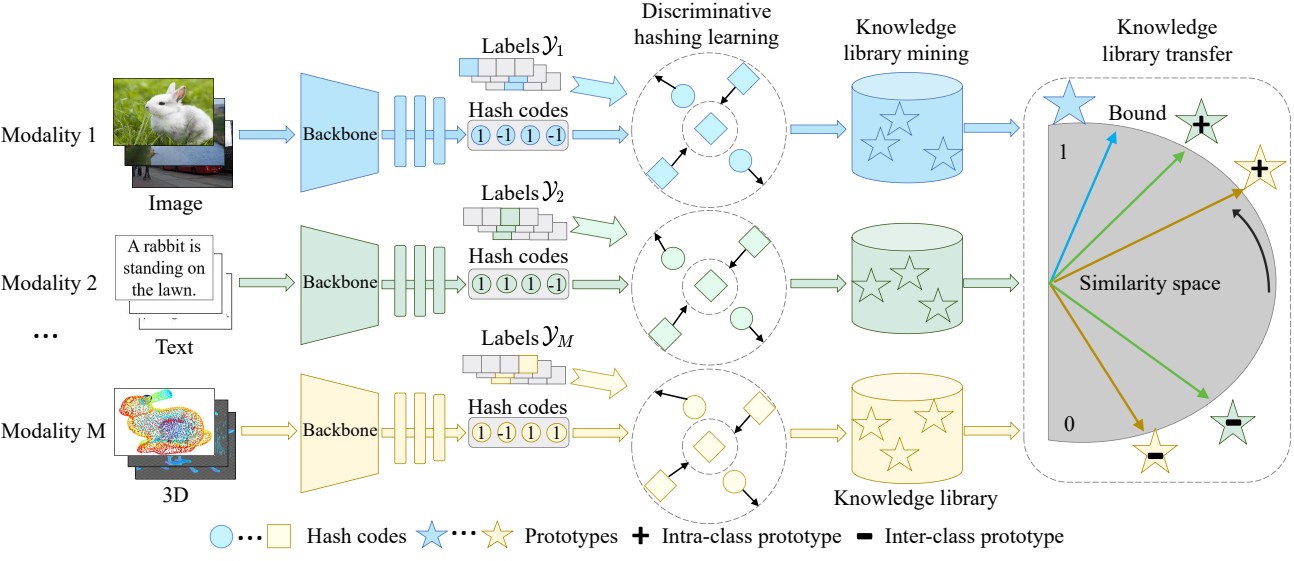

*Figure 1.* The framework of our SHE. Hash codes from different modalities are represented in blue, green, and yellow. Circles and rectangles represent different categories, while pentagrams denote class prototypes. Specifically, KLM extracts essential class prototypes, thereby explicitly constructing an implicit knowledge library for each modality. KLT aligns the knowledge library from the newly arrived modality with the historical knowledge library to maintain semantic consistency across modalities. DHL ensures the discriminability and compactness of hash codes by maximizing semantic similarity within the same class and semantic disparity across different classes.

undoubtedly increases the learning cost and limits the scalability and flexibility of CMR.

To enhance the flexibility of processing SMA, the CMR community has introduced a scalable paradigm (Wang & Peng, 2021; Pu et al., 2025a), which independently trains data from each modality. When new modality data arrives, the paradigm only requires training the sub-network specific to the new modality without retraining all sub-networks. For instance, MARS (Wang & Peng, 2021) leverages the shared label parsing module to achieve semantic alignment without interaction across modalities, thereby providing the fundamental conditions for independent training. Although these methods have achieved considerable performance, their low retrieval efficiency falls short of meeting the demands of large-scale data retrieval. Cross-modal hashing (CMH) has become an effective solution, due to its low storage requirements and high efficiency. However, how to construct the CMH framework to deal with SMA still remains an open research problem. In fact, this learning framework for SMA has the following key challenges: 1) Independent training of each modality in the CMH framework could amplify quantization errors, thereby leading to significant semantic information loss. 2) The absence of cross-modal semantic interaction hinders the learning of common semantic knowledge from SMA, thereby potentially degrading the quality of the generated hash codes. 3) When streaming data becomes unpaired, existing CMH methods fail to establish cross-modal semantic correspondences, thereby significantly reducing their effectiveness.

To address the above challenges, we propose a novel Streaming-media Hashing rEtrieval (SHE) method to achieve asynchronous retrieval for streaming-media data. Different from existing static CMH methods that require all modalities to be prepared before processing, our proposed SHE can learn hash codes of each modality in parallel to process streaming-media data without retraining all historical modalities, thus reducing the training complexity. The framework of our SHE is shown in Fig.1, which consists of a Knowledge Library Mining module (KLM), a Knowledge Library Transfer module (KLT), and a Discriminative Hashing Learning module (DHL). Specifically, we first propose the KLM module to extract essential class prototypes, thereby explicitly constructing an implicit knowledge library for each modality. Then, we propose the KLT module to align the knowledge of the newly arrived modality with the historical knowledge library, thereby preserving semantic consistency across modalities. Finally, we present DHL to learn compact and discriminative hash codes by maximizing the semantic relevance of intra-class samples and the semantic disparity of inter-class samples. In general, the major contributions of this work are shown as follows:

- This paper reveals and studies a practical but less-touched problem in cross-modal hashing, i.e., streaming-media hashing retrieval. To overcome this problem, we propose a novel cross-modal hashing

(CMH) retrieval framework dubbed streaming-media hashing retrieval (SHE). To the best of our knowledge, this could be the first work that addresses the problem of streaming-media data using a CMH paradigm.

- We propose a knowledge library mining module (KLM) and a knowledge library transfer (KLT) module that jointly extract an implicit knowledge library from the newcome data and align the commonality distribution from the new knowledge with ones from the historical knowledge library. This ensures that our SHE can process the newly arrived media data without retraining the whole historical data, thus reducing the training complexity.

- Extensive experiments of streaming-media retrieval on four widely used multi-modal datasets demonstrate the superiority and effectiveness of our proposed SHE compared with 14 state-of-the-art methods.

## 2. Related Work

Technically, cross-modal retrieval (CMR) aims at learning a series of modality-specific sub-networks to project multimodal data into a common space, enabling the direct measurement of similarity between data from different modalities (Sun et al., 2023; Duan et al., 2025; Pu et al., 2025b). Among these, cross-modal hashing (CMH), as an efficient and storage-friendly CMR technique, has garnered widespread attention from academia and industry. CMH methods are generally classified into unsupervised and supervised ones. Further, unsupervised ones (Qin et al., 2023; Liu et al., 2020; Shi et al., 2022) aim to capture the hidden semantic information by preserving the original structure of the multimodal data. For example, JDSH (Liu et al., 2020) constructs a joint-modal similarity matrix to maintain the multimodal semantic correlations among instances while incorporating a sampling and weighting mechanism to encourage the discriminative quality of hash codes. However, most existing unsupervised ones overly prioritize semantic consistency, overlooking the varying capacities of different hash functions to learn similarity. This imbalance often results in the hash function for one modality being weaker than those for others. To this end, DAEH (Shi et al., 2022) introduces an Information Mixed Similarity Estimation to estimate the similarity relations with discriminability while deploying an adaptive teacher guided enhancement optimization scheme to identify and strengthen weaker hash functions.

However, these methods still face performance bottlenecks due to the absence of semantic labels. In contrast, supervised methods (Qian et al., 2022; Qin et al., 2022; Wang et al., 2024c) enjoy a more considerable performance by leveraging semantic information in labels. For example, to leverage

the semantic multilevel advantages of the entire database and bridge the semantic and heterogeneity gaps between different modalities, MIAN (Zhang et al., 2022) explores asymmetric intra- and inter-modal similarity preservation under a probabilistic modality alignment framework. Unlike MIAN, which overlooks the relationship between the relative ranking of adjacent instances and the fine-grained label-level similarity, DCH-SCR (Liu et al., 2023) narrows the inherent gap between modalities by developing a ranking alignment loss function and introduces Normalized Discounted Cumulative Gain (NDCG) to achieve varying optimization intensities for data pairs with different similarities. Despite their decent performance, insufficient consideration is given to the fact that there is a spatial gap between real-number space and Hamming space, which may lead to solution space compression and loss function oscillation, thereby degrading performance. For this issue, SCH (Hu et al., 2024) divides sample pairs into fully semantically similar, partially semantically similar, and semantically dissimilar categories based on their similarity, and applies different constraints to each category to ensure the utilization of the entire Hamming space. Additionally, SCH introduces semantic channels to alleviate the issue of loss function oscillation.

The aforementioned CMH methods are implemented on the assumption that all media are simultaneously accessible, which is often impractical in real-world scenarios. When facing streaming-media data (SMD), i.e., the ever-emerging media, these methods must frequently retrain all modality-specific sub-networks, compromising the flexibility and scalability of CMR. In this regard, several real-valued CMR methods (Wang & Peng, 2021; Pu et al., 2025a) are proposed to allow for training only the newly added modality-specific network when new modality data arrives, greatly facilitating the flexibility of CMR. However, they are real-valued based representation methods, which involve high computational complexity and memory overhead, making it difficult to meet the fast retrieval demands of large-scale datasets. CMH, a technique celebrated for its high retrieval capabilities, presents a promising solution. However, how to design an effective CMH framework to deal with SMA remains an unresolved research gap. In this paper, we propose a novel CMH framework named streaming-media hashing retrieval(SHE), which integrates both high retrieval efficiency and flexibility in handling SMD.

## 3. The Proposed Method

### 3.1. Problem Formulation

Without loss of generality, given a multi-media dataset $\mathcal{D} = \{\mathcal{X}_m, \mathcal{Y}_m\}_{m=1}^{M}$ with $M$ modalities, where $\mathcal{X}_m$ and $\mathcal{Y}_m$ denote the data set and label set from the $m$-th modality respectively. Further, the data set and label set from the $m$-th modality are denoted as $\mathcal{X}_m = \{x_m^1, x_m^2, ..., x_m^{n_m}\}$

and $\mathcal{Y}_m = \{y_m^1, y_m^2, ..., y_m^{n_m}\}$ respectively, where $x_m^i$ denotes the $i$-th sample from the $m$-th modality, $y_m^i = \{y_m^{i,1}, y_m^{i,2}, ..., y_m^{i,C}\} \in \{0,1\}^{1 \times C}$ denotes the corresponding label, $C$ is the number of categories, and $n_m$ means the number of samples in $\mathcal{X}_m$. For a sample $x_m^i$, if it belongs to the $c$-th category, $y_m^{i,c} = 1$, otherwise $y_m^{i,c} = 0$. In our paper, the hash code of a sample $x_m^i$ is denoted as $b_m^i \in \{-1, 1\}^L$, where $L$ denotes the length of the binary code. Our main goal is to project $M$ modalities into the Hamming space to generate hash codes so that the similarity between different instances can be measured efficiently. To obtain discriminative hash codes, we first learn $M$ modality-specific sub-networks $\mathcal{H} = \{\mathcal{H}_m(\cdot, \Phi_m)\}_{m=1}^M$ for multi-media input. Note here that $\mathcal{H}_m$ represents the sub-network of the $m$-th modality and $\Phi_m$ denotes its corresponding learnable parameters. Then, the binary code of each sample $x_m^i$ could be generated by the sign function, i,e.,

$$b_m^i = \text{sign}(\mathcal{H}_m(x_m^i, \Phi_m)). \tag{1}$$

Considering that the sign function cannot be used for gradient-based optimization due to its non-differentiability, we replace the sign function with the tanh function to generate binary-like codes during training.

### 3.2. Overview

Different from traditional static learning methods that require all modalities to be prepared before processing, our SHE method can handle streaming-media data while multimedia data is continuously collected. In this paper, we propose a new streaming-media hashing retrieval (SHE) method, which is specifically designed to handle such streaming data. Since class prototypes could capture the consistency and specific information, we use them as the abstract knowledge library to preserve the instance commonality distribution of latent representations in streaming-media data. Moreover, such a library also alleviates the size pressure of historical multimedia data. Thus, the core of our SHE lies in extracting a prototype knowledge library that evolves over time. In general, SHE mainly consists of three modules, i.e., Knowledge Library Mining (KLM), Knowledge Library Transfer (KLT), and Discriminative Hashing Learning (DHL). To be specific, the KLM module constructs a knowledge library for every modality by extracting essential prototype knowledge, thereby serving as a semantic consistency maintenance medium. The KLT module aims to adaptively align the knowledge library extracted from the newly arrived modality with the historical knowledge library, thereby capturing the semantic consistency of multiple modalities. The DHL module focuses on reinforcing the compactness of intra-class samples and the separability of inter-class samples.

With the arrival of streaming-media data, the training pipeline of our SHE mainly contains two stages. In the

first stage, for the first available modality, we first employ KLM to extract an abstract knowledge library and push hash codes close to the corresponding class prototype knowledge. Then, we adopt DHL to learn high-quality hash codes. Thus, the objective loss could be formulated as

$$\mathcal{L}_m = \mathcal{L}_{klm} + \alpha \mathcal{L}_{klr} + \beta \mathcal{L}_{dhl}, \tag{2}$$

where $\alpha$ and $\beta$ are two hyper-parameters. $\mathcal{L}_{klm}$, $\mathcal{L}_{klr}$, and $\mathcal{L}_{dhl}$ represent the KLM loss, the KLR loss, and the DHL loss, respectively. In the second stage, for the new incoming media data, the knowledge library is updated by KLT to align the new knowledge with the historical knowledge library, thereby integrating the semantic information of the new modality into the knowledge library. Mathematically, the objective loss of the $m$-th modality ($m \geq 2$) can be written as

$$\mathcal{L}_m = \mathcal{L}_{klm} + \alpha(\mathcal{L}_{klr} + \mathcal{L}_{klt}) + \beta \mathcal{L}_{dhl}. \tag{3}$$

Therefore, the overall objective loss can be summarized as follows:

$$\mathcal{L} = \begin{cases} \mathcal{L}_{klm} + \alpha \mathcal{L}_{klr} + \beta \mathcal{L}_{dhl}, & \text{if } m = 1 \\ \mathcal{L}_{klm} + \alpha(\mathcal{L}_{klr} + \mathcal{L}_{klt}) + \beta \mathcal{L}_{dhl} & \text{if } m \geq 2 \end{cases}. \tag{4}$$

### 3.3. Knowledge Library Mining

Due to the absence of modality interaction, independently training modality-specific sub-networks for streaming media data is prone to semantic discrepancies. To this end, we utilize class prototypes to act as the abstract knowledge library, thereby preserving the semantic information of streaming data while alleviating the computational burden of maintaining historical data. When the data with a new modality arrives, we hope to extract the latent semantic information and convert it into a knowledge library, thereby building a bridge to reduce the heterogeneity gap between historical data and new data. Thus, we propose Knowledge Library Mining (KLM) to extract the abstract essential knowledge, thereby constructing a knowledge library for each modality as the medium. Specifically, we first utilize class prototypes as the abstract knowledge library. Considering the variability of intra-class samples, using only a single prototype for each class might result in a compromised representational capacity, potentially leading to suboptimal semantic alignment across modalities. Therefore, we assign multiple (i.e., $K$) prototypes to each class to better capture diverse semantic information, facilitating a more effective consistency among intra-class samples. Further, we randomly initialize and construct a learnable abstract knowledge library $P_m = \{p_m^1, p_m^2, ..., p_m^C\}$ for modality $m$ ($m \geq 1$), where $p_m^c = [p_m^{c,1}, p_m^{c,2}, ..., p_m^{c,K}]^T \in \mathbb{R}^{K \times L}$ denotes $K$ normalized and binary prototypes for the $c$-th class. Then, we encourage hash codes and their closest homo-category

prototype to move closer to each other. To this end, we denote the similarity between any sample $x_m^i$ from the $m$-th modality and its $k$-th corresponding prototype as follows,

$$s(x_m^i|k) = \frac{\sum_{c=1}^C y_m^{i,c} \cdot \Gamma(b_m^i, p_m^{c,k}) + 1}{2}, \qquad (5)$$

where $\Gamma(\cdot, \cdot)$ is the cosine similarity function. Thus, the loss function could be formalized as follows:

$$\mathcal{L}_{klm} = -\frac{1}{n_m} \sum_{i=1}^{n_m} \log \left( \max_{k=1,...,K} s\left(x_m^i|k\right) \right). \qquad (6)$$

The above loss aims to encourage each training sample to be close to the nearest prototype with the same category. However, this KLM loss could lead to a trivial solution where all prototypes collapse into one point, thus weakening the discriminative ability of hash codes. To this end, we develop a Knowledge Library Regularization (KLR) mechanism to enhance the distinctiveness between inter-class prototypes while ensuring semantic consistency between intra-class prototypes. The KLR loss can be written as:

$$\mathcal{L}_{klr} = -\frac{1}{CK} \sum_{c=1}^C \sum_{k=1}^K \log \frac{\sum_{k'=1,k'\neq k}^K e^{\Gamma(p_m^{c,k}, p_m^{c,k'})}}{\sum_{\hat{c}=1}^C \sum_{\hat{k}=1}^K e^{\Gamma(p_m^{c,k}, p_m^{\hat{c},\hat{k}})} - e}. \qquad (7)$$

### 3.4. Knowledge Library Transfer

Since streaming media data is constantly changing and updating, the model needs to have good generalization ability to adapt to such changes. To this end, we propose Knowledge Library Transfer (KLT) to extract knowledge from new media data and transfer its semantic information to the benchmark knowledge library. This allows our SHE to process new media without retraining the entire historical media data, thereby reducing computational complexity while ensuring semantic consistency between streaming modal data. Therefore, for the modality $m$ $(m \geq 2)$, the KLT loss is defined as follows, i.e.,

$$\mathcal{L}_{klt} = -\frac{1}{CK} \sum_{c=1}^C \sum_{k=1}^K \log \left[ min(1, max(0, \Gamma(p_1^{c,k}, p_m^{c,k}) - \sigma + 1)) \right], \qquad (8)$$

where $\sigma$ is the similarity boundary.

### 3.5. Discriminative Hashing Learning

To strengthen the compactness of intra-class samples and the scatter of inter-class samples, we employ a Discriminative Hashing Learning (DHL) module to constrain the learning of all sub-networks. Specifically, DHL strives to maximize the similarity of hash codes within the same class while minimizing the similarity of hash codes from different classes. For modality $m$ $(m \geq 1)$, the DHL loss could be expressed as the following formula, i.e.,

$$\mathcal{L}_{dhl} = -\frac{1}{n_m} \sum_{i=1}^{n_m} \log \frac{\sum_{j=1}^{n_m} \Theta_m^{i,j} \Gamma(b_m^i, b_m^j)}{\sum_{j=1}^{n_m} \Gamma(b_m^i, b_m^j)}, \qquad (9)$$

where $\Theta_m^{i,j}$ is an indicator that measures the similarity between two samples, and its value is 1 if $x_m^i$ and $x_m^j$ are intra-class samples, otherwise 0.

### 3.6. Optimization

To learn the optimal hashing functions, we jointly minimize the above losses. Specifically, for the $m$-th modality, the overall loss function is defined as follows:

$$\mathcal{L} = \begin{cases} \mathcal{L}_{klm} + \alpha\mathcal{L}_{klr} + \beta\mathcal{L}_{dhl}, & \text{if } m = 1 \\ \mathcal{L}_{klm} + \alpha(\mathcal{L}_{klr} + \mathcal{L}_{klt}) + \beta\mathcal{L}_{dhl} & \text{if } m \geq 2 \end{cases}. \qquad (10)$$

To offer a comprehensive overview of our SHE framework, we summarize the training process in Algorithm 1, which outlines the optimization pipeline across modalities.

## 4. Experiments

### 4.1. Dataset

In our experiment, we evaluate the proposed SHE on four widely used multimedia datasets, namely, Wikipedia (Rasiwasia et al., 2010), NUS-WIDE (Chua et al., 2009), XMedia (Peng et al., 2015), and XMediaNet (Peng et al., 2018). In Tab.1, we summarize the statistics of the datasets. Notably, to align with the streaming-media scenario, we gradually incorporate the modalities into the training process in the order they are collected within the datasets. More details about the four datasets are presented in the appendix.

Table 1. General statistics of the four datasets, where "*/*/*" in the "Instance" column indicates the number of samples for training, testing, and retrieval database, respectively.

| Dataset | Modality | Instance | Feature |
|---|---|---|---|
| Wikipedia | Image | 2,173/693/2,866 | 4,096d VGG |
| | Text | 2,173/693/2,866 | 300d Doc2Vec |
| NUS-WIDE | Image | 8,000/2,000/10,000 | 4,096d VGG |
| | Text | 8,000/2,000/10,000 | 300d Doc2Vec |
| XMedia | Image | 4,000/1,000/5,000 | 4,096d VGG |
| | Text | 4,000/1,000/5,000 | 3,000d Bow |
| | Audio clip | 800/200/1,000 | 29d MFCC |
| | 3D | 400/100/500 | 4,700d LightField |
| | Video | 969/174/1,143 | 4,096d C3D |
| XMediaNet | Image | 32,000/8,000/40,000 | 4,096d VGG |
| | Text | 32,000/8,000/40,000 | 300d Doc2Vec |
| | Audio clip | 8,000/2,000/10,000 | 672d MFCC |
| | 3D | 1,600/400/2,000 | 4,700d LightField |
| | Video | 8,000/2,000/10,000 | 4,096d C3D |

### 4.2. Experimental Settings and Evaluation Metric

In our SHE, all modality-specific sub-networks comprise three fully connected layers, with the ReLU activation function applied after the first two layers and a $\ell_2$-normalization operation applied to the last layer. Their dimensions are $[d^* \rightarrow 4096 \rightarrow 4096 \rightarrow L]$, where $d^*$ represents the input feature dimensions of the corresponding modality. For all datasets, we set the batch size $n_b$ as 256, the similarity boundary $\sigma$ as 0.95, the iteration number $T$

---

**Algorithm 1** The training process of our SHE

---

**Input:** Multi-modal data $\mathcal{D} = \{\mathcal{X}_m, \mathcal{Y}_m\}_{m=1}^M$, $\alpha$, $\beta$, the number of prototypes $K$, the similarity boundary $\sigma$, sub-networks $\mathcal{H} = \{\mathcal{H}_m(\cdot, \Phi_m)\}_{m=1}^M$, hash length $L$, batch size $n_b$, learning rate $l_r$, and iteration number $T$.

// The first stage (m = 1)

1: Randomly generate a learnable abstract knowledge library $P_1 = \{p_1^1, p_1^2, ..., p_1^C\}$.
2: Randomly initialize $\Phi_1$.
3: **for** $iter = 1, 2, \ldots,$ **to** $T$ **do**
4:   **for** $step = 1, 2, \ldots,$ **to** $\left\lfloor \frac{n_1}{n_b} \right\rfloor$ **do**
5:     Get a mini-batch $(\widetilde{\mathcal{X}}_1, \widetilde{\mathcal{Y}}_1)$ from $(\mathcal{X}_1, \mathcal{Y}_1)$.
6:     Compute hash codes for $\widetilde{\mathcal{X}}_1$ by the sub-network $\mathcal{H} = \{\mathcal{H}_1(\cdot, \Phi_1)\}$.
7:     Update the knowledge library $P_1$ by $\tanh$ function and normalization.
8:     Compute the loss $\mathcal{L}$ according to Eq.10.
9:     Compute gradients of $\Phi_1$ and $P_1$ and update them:
       $\Phi_1 = \Phi_1 - l_r \frac{\partial \mathcal{L}}{\partial \Phi_1}$
       $P_1 = P_1 - l_r \frac{\partial \mathcal{L}}{\partial P_1}$.
10:   **end for**
11: **end for**

// The second stage (m ≥ 2)

1: Use the mined knowledge library $CK_1$ to steer the semantic alignment for subsequent modalities.
2: **for** $m = 2, 3, \ldots,$ **to** $M$ **do**
3:   Randomly generate a learnable abstract knowledge library $P_m = \{P_m^1, P_m^2, ..., P_m^C\}$.
4:   Randomly initialize $\Phi_m$.
5:   **for** $iter = 1, 2, \ldots,$ **to** $T$ **do**
6:     **for** $step = 1, 2, \ldots,$ **to** $\left\lfloor \frac{n_m}{n_b} \right\rfloor$ **do**
7:       Get a mini-batch $(\widetilde{\mathcal{X}}_m, \widetilde{\mathcal{Y}}_m)$ from $(\mathcal{X}_m, \mathcal{Y}_m)$.
8:       Compute hash codes for $\widetilde{\mathcal{X}}_1$ by the sub-network $\mathcal{H} = \{\mathcal{H}_m(\cdot, \Phi_m)\}$.
9:       Update the knowledge library $P_m$ by $\tanh$ function and normalization.
10:      Compute the loss $\mathcal{L}$ according to Eq.10.
11:      Compute gradients of $\Phi_m$ and $P_m$ and update them:
        $\Phi_m = \Phi_m - l_r \frac{\partial \hat{\mathcal{L}}}{\partial \Phi_m}$
        $P_m = P_m - l_r \frac{\partial \hat{\mathcal{L}}}{\partial P_m}$.
12:    **end for**
13:  **end for**
14: **end for**

**Output:** The optimized parameters $\{\Phi_m\}_{m=1}^M$.

---

as 300, the number of prototype vectors $K$ as 3, and the hyperparameter $\alpha$ as 1. For four datasets, we set the hyperparameter $\beta$ as 4, 5, 1, and 6, respectively. Additionally, the SHE framework is implemented by the PyTorch toolkit,

and all experiments are conducted on a single GeForce RTX3090Ti 24GB GPU.

Following previous works (Wang & Peng, 2021; Pu et al., 2025a), we evaluate the performance of SHE by performing cross-modal retrieval (CMR) tasks, i.e., retrieving one modality using queries from another modality. For instance, we designate images as queries to retrieve text (i.e., I2T) and text as queries to retrieve images(i.e., T2I). Besides, the Mean Average Precision (MAP) score, a widely adopted metric in the CMR community, is employed as the evaluation metric in our experiment since it simultaneously reflects both the precision and recall of the retrieved results.

### 4.3. Comparison methods

To demonstrate the superiority of the proposed SHE, we compare it against 14 state-of-the-art (SOTA) methods, including three real-valued methods (i.e., MARS (Wang & Peng, 2021), GNN4CMR (Qian et al., 2022), and HOPE (Zhang et al., 2024) ), three unsupervised hashing methods (i.e., DGCPN (Yu et al., 2021), CIRH (Zhu et al., 2022), and UCCH (Hu et al., 2022)), and eight supervised hashing methods (i.e., HMAH (Tan et al., 2022), CMMQ (Yang et al., 2022), MIAN (Zhang et al., 2023), DSPH (Huo et al., 2023), DHRL (Shu et al., 2024), DHaPH (Huo et al., 2024), DNpH (Qin et al., 2024), and SCH (Hu et al., 2024)). All methods except MARS and our SHE are limited to training on paired data from two modalities, and we refer to these methods as paired-oriented methods in the section. When paired-oriented methods face datasets with five modalities, we first construct pseudo-instance pairs through label-based repeat sampling and then conduct training on every two modalities, resulting in a total of $5 \times 4/2$ training processes. For the Wikipedia and NUS-WIDE datasets, we only compare hashing methods. For the XMedia and XMediaNet datasets, we compare both hashing methods and real-valued methods.

### 4.4. Comparisons with State-of-the-Art Methods

**Quantitative Comparisons:** To demonstrate the superiority of our SHE, we compare it with SOTA methods in terms of MAP scores. Tab.2 reports the comparison between our SHE and SOTA methods on the Wikipedia and NUS-WIDE datasets with different bit lengths (i.e., 16, 32, 64, and 128). For the XMedia and XMediaNet datasets, we only present experimental results with 128 bits in Tab.3 and Tab.4, and a detailed analysis of the impact of bit length on performance is provided in the appendix. From these results, it can be concluded: 1) Due to the lack of label supervision, unsupervised methods suffer from a compromise in performance, compared with supervised ones. 2) The XMediaNet dataset, with more categories (200), presents increased learning difficulty, leading to a significant performance decline across all

Table 2. The MAP scores with different bit lengths on the Wikipedia and NUS-WIDE datasets.

| Method | Ref. | Wikipedia | | | | | | | | | | NUS-WIDE | | | | | | | | | |
| --- | --- | --- | --- | --- | --- | --- | --- | --- | --- | --- | --- | --- | --- | --- | --- | --- | --- | --- | --- | --- | --- |
| | | I2T | | | | | T2I | | | | | I2T | | | | | T2I | | | | |
| | | 16 | 32 | 64 | 128 | Avg | 16 | 32 | 64 | 128 | Avg | 16 | 32 | 64 | 128 | Avg | 16 | 32 | 64 | 128 | Avg |
| DGCPN | AAAI'21 | 21.6 | 23.8 | 29.0 | 29.9 | 26.1 | 21.7 | 24.7 | 30.5 | 30.4 | 26.8 | 50.9 | 50.8 | 51.8 | 52.3 | 51.5 | 48.7 | 49.3 | 50.3 | 51.1 | 49.9 |
| CIRH | TKDE'22 | 29.1 | 31.0 | 29.9 | 25.0 | 28.8 | 27.8 | 31.3 | 28.5 | 24.0 | 27.9 | 41.8 | 45.4 | 46.0 | 46.6 | 45.0 | 43.0 | 47.1 | 47.4 | 47.9 | 46.4 |
| UCCH | TPAMI'23 | 25.7 | 29.3 | 30.9 | 32.6 | 29.6 | 24.7 | 28.9 | 30.3 | 33.2 | 29.3 | 42.9 | 44.9 | 48.2 | 49.4 | 46.4 | 42.2 | 44.0 | 47.8 | 48.5 | 45.6 |
| HMAH | TMM'22 | 50.2 | 51.9 | 51.7 | 51.2 | 51.3 | 69.9 | 74.0 | 75.7 | 76.2 | 74.0 | 59.7 | 61.0 | 61.9 | 61.9 | 61.1 | 64.4 | 66.7 | 67.8 | 67.8 | 66.7 |
| CMMQ | CVPR'22 | 55.8 | 55.6 | 56.5 | 56.2 | 56.0 | 66.5 | 72.6 | 75.7 | 76.0 | 72.7 | 58.3 | 59.2 | 59.5 | 59.5 | 59.1 | 63.3 | 66.0 | 66.1 | 65.2 | 65.2 |
| MIAN | TKDE'23 | 46.1 | 43.7 | 45.6 | 47.4 | 45.7 | 38.2 | 38.6 | 39.6 | 43.5 | 40.0 | 60.3 | 62.8 | 62.1 | 62.8 | 62.0 | 47.9 | 52.6 | 54.3 | 56.5 | 52.8 |
| DSPH | TCSVT'23 | 45.3 | 46.2 | 48.2 | 48.7 | 47.1 | 52.0 | 59.4 | 59.1 | 57.0 | 56.9 | 56.1 | 57.7 | 58.7 | 58.2 | 57.7 | 63.0 | 64.5 | 65.1 | 65.8 | 64.6 |
| DHRL | TBD'24 | 46.8 | 48.2 | 50.7 | 51.5 | 49.3 | 46.9 | 47.1 | 52.3 | 51.6 | 49.5 | 55.1 | 64.6 | 60.5 | 64.3 | 61.1 | 54.8 | 61.2 | 61.4 | 62.2 | 59.9 |
| DHaPH | TKDE'24 | 45.7 | 48.6 | 51.0 | 51.7 | 49.3 | 59.7 | 59.7 | 65.9 | 61.6 | 61.7 | 56.6 | 58.8 | 60.3 | 60.2 | 59.0 | 63.0 | 63.3 | 64.6 | 64.1 | 63.8 |
| DNpH | TMM'24 | 46.8 | 48.2 | 50.6 | 50.9 | 49.1 | 62.1 | 58.4 | 64.2 | 56.7 | 60.4 | 55.9 | 58.4 | 59.2 | 59.6 | 58.3 | 63.5 | 65.5 | 65.3 | 65.7 | 65.0 |
| SCH | TPAMI'24 | 51.0 | 55.4 | 57.1 | 56.3 | 55.0 | 67.1 | 74.0 | 76.4 | 77.2 | 73.7 | 60.1 | 60.3 | 62.2 | 61.6 | 61.1 | 64.6 | 65.3 | 66.6 | 65.9 | 65.6 |
| SHE | Ours | **56.3** | **58.6** | **59.5** | **59.4** | **58.5** | **71.7** | **74.6** | **77.7** | **77.9** | **75.5** | **67.0** | **68.0** | **70.7** | **69.8** | **68.9** | **67.2** | 65.9 | **69.4** | **69.2** | **67.9** |

Table 3. The MAP scores with 128 bits on the XMedia dataset.

| Methods | Ref. | XMedia | | | | | | | | | | | | | | | | | | | | Avg |
| --- | --- | --- | --- | --- | --- | --- | --- | --- | --- | --- | --- | --- | --- | --- | --- | --- | --- | --- | --- | --- | --- | --- |
| | | Image | | | | Text | | | | Audio | | | | 3D | | | | Video | | | | |
| | | Text | Audio | 3D | Video | Image | Audio | 3D | Video | Image | Text | 3D | Video | Image | Text | Audio | Video | Image | Text | Audio | 3D | |
| DGCPN | AAAI'21 | 57.0 | 30.9 | 10.1 | 80.0 | 45.8 | 28.2 | 7.1 | 33.1 | 24.5 | 32.3 | 6.0 | 6.7 | 7.7 | 6.2 | 5.7 | 5.7 | 56.3 | 30.6 | 5.6 | 6.1 | 24.3 |
| CIRH | TKDE'22 | 85.9 | 11.6 | 51.3 | 74.4 | 86.3 | 8.4 | 35.7 | 52.2 | 11.0 | 8.2 | 9.8 | 11.6 | 44.5 | 35.8 | 10.0 | 29.0 | 50.7 | 42.3 | 8.4 | 24.0 | 34.6 |
| UCCH | TPAMI'23 | 84.0 | 31.0 | 29.3 | 75.0 | 88.4 | 22.8 | 9.9 | 56.4 | 27.0 | 23.8 | 12.2 | 19.0 | 14.8 | 10.1 | 6.8 | 4.9 | 52.8 | 43.6 | 14.6 | 7.3 | 31.7 |
| HMAH | TMM'22 | 91.4 | 53.9 | 87.3 | 86.5 | 95.8 | 57.6 | 92.0 | 91.4 | 54.2 | 52.4 | 40.4 | 29.1 | 61.6 | 63.8 | 30.1 | 57.4 | 57.9 | 56.0 | 19.1 | 52.6 | 61.5 |
| CMMQ | CVPR'22 | 91.4 | 26.4 | 87.5 | 83.9 | 94.9 | 9.8 | 31.2 | 30.8 | 29.5 | 10.7 | 12.1 | 8.5 | 69.1 | 27.1 | 7.6 | 34.0 | 47.6 | 19.0 | 6.4 | 23.4 | 37.5 |
| MIAN | TKDE'23 | 85.2 | 29.0 | 80.2 | 76.8 | 88.0 | 6.5 | 9.7 | 9.5 | 24.5 | 5.2 | 8.4 | 9.2 | 56.7 | 5.2 | 5.8 | 5.8 | 51.8 | 5.3 | 5.8 | 6.1 | 28.7 |
| DSPH | TCSVT'23 | 91.7 | 41.7 | 24.5 | 86.0 | 95.8 | 41.4 | 20.1 | 90.8 | 50.6 | 47.8 | 8.9 | 21.4 | 14.9 | 10.3 | 7.0 | 6.4 | 56.4 | 57.0 | 14.6 | 11.1 | 39.9 |
| DHRL | TBD'24 | 90.1 | 74.9 | 81.6 | 84.3 | 94.4 | 81.6 | 86.5 | 84.1 | 60.0 | 61.9 | 45.6 | 37.0 | 61.8 | 63.5 | 43.3 | 45.1 | 56.9 | 51.7 | 29.0 | 38.9 | 63.6 |
| DHaPH | TKDE'24 | 91.9 | 45.2 | 48.5 | 86.3 | 96.2 | 44.7 | 40.6 | 91.4 | 52.3 | 48.5 | 11.7 | 33.7 | 43.8 | 35.3 | 8.1 | 19.8 | 60.2 | 59.6 | 17.4 | 16.8 | 47.6 |
| DNpH | TMM'24 | 91.8 | 36.9 | 36.8 | 85.4 | 96.2 | 40.0 | 33.6 | 91.5 | 47.4 | 46.2 | 12.1 | 27.8 | 43.2 | 37.3 | 10.6 | 10.2 | 58.0 | 59.8 | 16.2 | 15.2 | 44.8 |
| SCH | TPAMI'24 | 90.1 | 85.9 | 80.5 | 87.2 | 94.4 | 62.2 | 52.6 | 88.3 | 64.4 | 45.2 | 12.8 | 28.7 | 66.0 | 44.8 | 15.3 | 8.9 | 59.7 | 54.9 | 20.0 | 9.6 | 53.6 |
| MARS | TCSVT'22 | 91.4 | 84.3 | 88.3 | 85.5 | 92.9 | 85.8 | 90.2 | 87.3 | 63.5 | 64.3 | 61.4 | 57.0 | 70.8 | 71.2 | 65.2 | 65.6 | 55.8 | 55.5 | 50.3 | 53.3 | 72.0 |
| GNN4CMR | TPAMI'23 | 90.9 | 40.7 | 86.8 | 84.9 | 92.2 | 36.4 | 87.2 | 87.6 | 46.8 | 44.9 | 40.0 | 37.9 | 62.6 | 62.8 | 26.0 | 57.0 | 54.3 | 55.5 | 19.5 | 47.8 | 58.1 |
| HOPE | TPAMI'24 | 91.0 | 71.1 | 86.8 | 84.8 | 95.4 | 66.2 | 87.3 | 82.7 | 59.4 | 54.0 | 47.6 | 40.3 | 63.5 | 61.8 | 41.7 | 49.9 | 53.5 | 51.4 | 27.0 | 36.6 | 62.6 |
| SHE | Ours | **92.1** | **87.3** | **89.6** | **87.2** | 95.8 | **92.0** | **94.0** | **91.7** | 65.3 | **66.5** | **64.8** | **62.3** | 70.2 | 70.4 | **67.7** | **66.4** | 60.7 | 61.3 | **57.2** | **58.4** | **75.0** |

Table 4. The MAP scores with 128 bits on the XMediaNet dataset, where '/' means out of memory.

| Methods | Ref. | XMediaNet | | | | | | | | | | | | | | | | | | | | Avg |
| --- | --- | --- | --- | --- | --- | --- | --- | --- | --- | --- | --- | --- | --- | --- | --- | --- | --- | --- | --- | --- | --- | --- |
| | | Image | | | | Text | | | | Audio | | | | 3D | | | | Video | | | | |
| | | Text | Audio | 3D | Video | Image | Audio | 3D | Video | Image | Text | 3D | Video | Image | Text | Audio | Video | Image | Text | Audio | 3D | |
| DGCPN | AAAI'21 | / | / | / | / | / | / | / | / | / | / | / | / | / | / | / | / | / | / | / | / | / |
| CIRH | TKDE'22 | / | / | / | / | / | / | / | / | / | / | / | / | / | / | / | / | / | / | / | / | / |
| UCCH | TPAMI'23 | 32.6 | 35.1 | 10.1 | 47.1 | 25.1 | 8.8 | 3.7 | 14.5 | 22.8 | 8.8 | 2.7 | 0.8 | 2.0 | 1.5 | 1.4 | 1.2 | 30.0 | 13.9 | 0.8 | 1.5 | 13.2 |
| HMAH | TMM'22 | 44.2 | 59.6 | 70.2 | 72.0 | 42.5 | 5.1 | 29.6 | 19.1 | 33.1 | 7.3 | 19.4 | 0.6 | 37.6 | 24.4 | 12.7 | 31.6 | 39.4 | 18.4 | 0.6 | 34.5 | 30.1 |
| CMMQ | CVPR'22 | / | / | / | / | / | / | / | / | / | / | / | / | / | / | / | / | / | / | / | / | / |
| MIAN | TKDE'23 | / | / | / | / | / | / | / | / | / | / | / | / | / | / | / | / | / | / | / | / | / |
| DSPH | TCSVT'23 | 18.3 | 23.8 | 14.1 | 52.9 | 33.4 | 4.3 | 4.1 | 12.4 | 26.7 | 3.1 | 1.5 | 0.6 | 3.3 | 1.1 | 0.7 | 1.5 | 35.0 | 6.9 | 0.9 | 1.7 | 12.3 |
| DHRL | TBD'24 | 54.3 | 55.9 | 64.4 | 61.7 | 40.5 | 33.1 | 35.8 | 37.4 | 38.2 | 26.9 | 25.4 | 0.7 | 33.3 | 25.0 | 14.6 | 25.2 | 32.9 | 24.7 | 0.9 | 26.3 | 32.9 |
| DHaPH | TKDE'24 | 29.0 | 27.3 | 32.3 | 60.9 | 42.0 | 5.1 | 6.6 | 22.4 | 27.2 | 3.6 | 2.2 | 0.7 | 18.8 | 2.3 | 1.0 | 5.6 | 39.3 | 12.3 | 0.6 | 4.2 | 17.2 |
| DNpH | TMM'24 | 18.4 | 21.7 | 27.5 | 44.4 | 31.8 | 4.8 | 6.4 | 16.2 | 26.8 | 3.6 | 2.4 | 0.8 | 19.7 | 2.4 | 1.3 | 4.7 | 34.3 | 8.5 | 0.7 | 4.5 | 14.0 |
| SCH | TPAMI'24 | 67.4 | 42.4 | 56.6 | **75.6** | 66.0 | 3.7 | 12.0 | 23.3 | 35.1 | 1.3 | 2.1 | 0.7 | 33.3 | 10.4 | 1.7 | 6.5 | **46.8** | 15.6 | 0.6 | 5.4 | 25.3 |
| MARS | TCSVT'22 | 70.0 | 67.5 | 67.3 | 61.6 | 63.9 | 57.2 | 57.4 | 51.1 | 40.2 | 36.7 | 35.9 | 30.6 | 40.8 | 37.2 | 35.0 | 31.7 | 37.4 | 34.0 | 32.4 | 33.2 | 46.1 |
| GNN4CMR | TPAMI'23 | 47.0 | 51.6 | 70.2 | 73.3 | 61.3 | 12.3 | 52.4 | 49.4 | 36.1 | 8.9 | 17.2 | 0.7 | 36.7 | 20.7 | 7.6 | 25.2 | 43.5 | 22.7 | 0.8 | 30.8 | 33.4 |
| HOPE | TPAMI'24 | 51.8 | 58.3 | 67.5 | 68.2 | 54.0 | 43.8 | 56.2 | 57.0 | 45.0 | 39.2 | 32.3 | 0.6 | 38.7 | 35.2 | 18.7 | 21.1 | 40.1 | 35.9 | 1.1 | 23.8 | 39.4 |
| SHE | Ours | **75.1** | **70.6** | **70.3** | 72.8 | **70.5** | **62.8** | **63.5** | **65.8** | 42.2 | **39.8** | **36.7** | **38.2** | **41.2** | **39.3** | 34.0 | **37.7** | 43.1 | **41.0** | **36.4** | **38.3** | **51.0** |

methods. 3) Paired-oriented methods typically rely on true-instance pairs to bridge the cross-modal gap, which hinders their performance on unpaired datasets (i.e., the XMedia and XMediaNet datasets), resulting in performance inferior

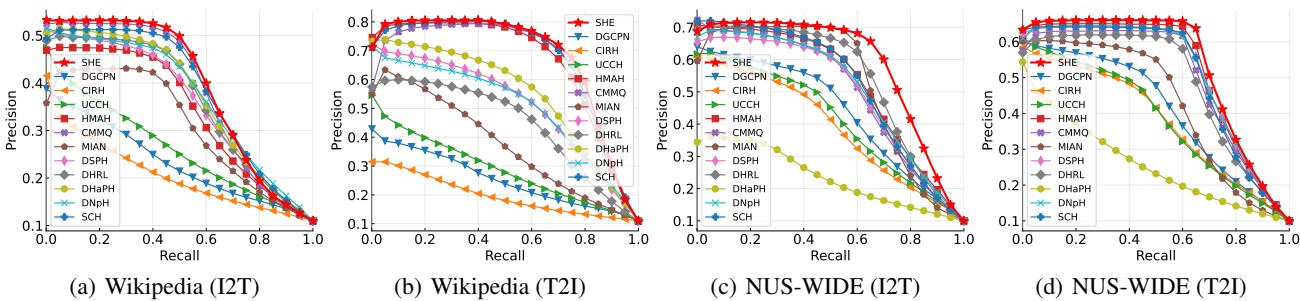

*Figure 2.* Precision-recall curves with 128 bits on the Wikipedia and NUS-WIDE datasets.

to MARS and our SHE. 4) MARS simply deploys a shared label parsing module to guide the learning process across all modalities. In comparison, our SHE outperforms them by mining knowledge libraries, regularizing knowledge libraries, and aligning knowledge libraries for consistency. This holistic strategy greatly enhances the quality of the hash codes, resulting in performance that surpasses that of MARS.

**Qualitative Comparisons:** To further evaluate the retrieval performance of SHE, we compare it with SOTA methods in terms of precision-recall curves (PR-curves). Fig.2 plots PR-curves with 128 bits on the Wikipedia and NUS-WIDE datasets. As shown in the figure, our proposed SHE consistently achieves higher curves compared to SOTA methods, indicating superior precision at the same recall levels. This further confirms the superiority of SHE.

### 4.5. Ablation Study

To verify the effectiveness of our components, we conduct an ablation study with 128 bits on four datasets and report the corresponding mean MAP scores of all cross-modal retrieval tasks. The results are presented in Tab.5, where **SHE w/o KLM** represents the removal of KLM loss, **SHE w/o KLR** indicates the exclusion of KLR loss, **SHE w/o KLT** denotes the deletion of KLT loss, and **SHE w/o DHL** refers to the absence of DHL loss. From these results, we can obtain the following findings: 1) The absence of KLM suffers a significant performance degradation, likely because KLM is capable of extracting essential prototype knowledge and building effective knowledge libraries. 2) The performance degradation without KLR could be attributed to the fact that KLR can enhance the discrimination of inter-class prototypes in the knowledge library, which is beneficial for learning high-quality hash codes. 3) The removal of KLT leads to a performance decline. This is likely due to the ability of KLT to align knowledge libraries mined on different modalities, thereby preserving semantic consistency across modalities. 4) The exclusion of DHL similarly results in lower performance, likely due to the ability of

DHL to enhance the discriminability of inter-class samples and the compactness of intra-class samples. In conclusion, the removal of any single component leads to performance degradation, highlighting the importance of each proposed component in addressing the cross-modal retrieval problem in streaming-media scenarios.

*Table 5.* Ablation study with 128 bits.

|  | Wikipedia | NUS-WIDE | XMedia | XMediaNet |
|---|---|---|---|---|
| SHE w/o KLM | 21.6 | 12.2 | 13.4 | 0.9 |
| SHE w/o KLR | 54.3 | 68.2 | 73.8 | 33.3 |
| SHE w/o KLT | 68.0 | 69.0 | 39.8 | 29.0 |
| SHE w/o DHL | 56.7 | 64.3 | 38.2 | 31.4 |
| SHE | **68.6** | **69.5** | **75.0** | **51.0** |

### 4.6. Parameter Analysis

To investigate the sensitivity of two parameters (i.e., $\alpha$ and $\beta$), we conduct experiments with 128 bits on the XMedia and XMediaNet datasets. Specifically, we try one parameter with different values while keeping another parameter fixed to evaluate its individual impact on performance. As presented in Fig.3, the performance of our proposed SHE method on both datasets initially improves as the parameter value increases, and then either stabilizes or slightly declines. These results demonstrate that our method maintains decent performance across a relatively wide range of parameter values. In general, the performance is more stable when $\alpha$ is set between 4 and 8, and the best results are achieved when $\beta$ is set to 1.

### 4.7. The Impact of Media Learning Sequence

To delve into the impact of media learning sequences with 128 bits on the XMedia and XMediaNet datasets, we sequentially utilize the Image, Text, Audio, 3D, and Video modalities as the first available modality for knowledge library mining, with the corresponding mean MAP scores of all cross-modal retrieval tasks presented in Fig.4. The results show that for the XMediaNet dataset, the performance

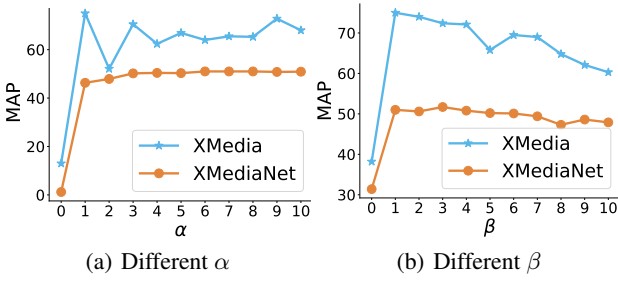

(a) Different $\alpha$        (b) Different $\beta$

*Figure 3.* Parameter analysis with 128 bits on the XMedia and XMediaNet datasets.

remains relatively stable regardless of the choice of the first available modality. However, for the XMedia dataset, while the performance is similarly stable across most modality choices, it exhibits a notable decline for the Audio modality. This discrepancy is likely attributable to the reliance of the Audio modality in the XMedia dataset on low-dimensional MFCC features (i.e., 29-dimensional MFCC), which lack sufficient representational capacity. Consequently, this limitation results in a lower-quality knowledge library, ultimately leading to significantly degraded performance.

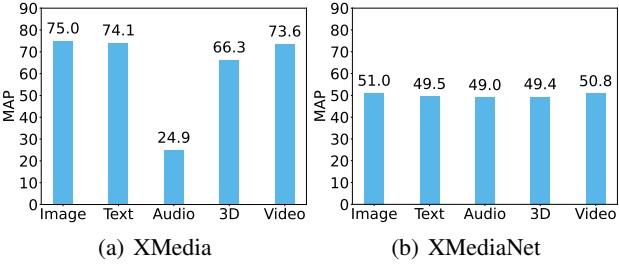

(a) XMedia        (b) XMediaNet

*Figure 4.* The mean MAP scores of all cross-modal retrieval tasks with 128 bits for different choices of the first available modality on the XMedia and XMediaNet datasets.

### 4.8. The Impact of Similarity Boundary

In our KLT, for intra-class prototypes across different modalities, we focus on maintaining a certain level of semantic similarity between them by setting a similarity boundary rather than forcing them to be identical. To investigate the impact of different similarity boundaries on performance and verify the effectiveness of the above strategy, we conduct experiments with 128 bits on the XMedia dataset by varying the similarity boundary $\sigma$ and report the corresponding mean MAP scores of all cross-modal retrieval tasks. As presented in Fig.5(a), the performance follows a single peak trend as $\sigma$ increases, reaching the optimal result at $\sigma = 0.95$. The finding highlights that maintaining a certain level of semantic similarity is more effective in guiding the learning of different modalities compared to directly enforcing

identical prototype knowledge (i.e., $\sigma = 1.0$).

### 4.9. The Impact of Knowledge Library Scale

The number of class prototypes $K$ can influence the scale of the knowledge library, which may in turn impact the quality of the mined prototype knowledge. To investigate the impact of the knowledge library scale on performance, we perform experiments with 128 bits on the XMedia dataset by varying $K$. As depicted in Fig.5(b), when $K$ takes most values (i.e., 3, 5, and 9), the performance consistently surpasses that at $K = 1$. This observation underscores the advantage of assigning multiple prototypes to each category, as it can alleviate semantic information loss, strengthen semantic consistency, and ultimately enhance overall performance.

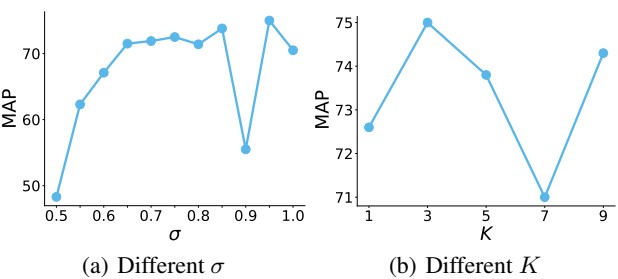

(a) Different $\sigma$        (b) Different $K$

*Figure 5.* The impact of similarity boundary and knowledge library scale on the XMedia dataset. a) The mean MAP scores of all cross-modal retrieval tasks with 128 bits under different similarity boundaries $\sigma$. b) The mean MAP scores of all cross-modal retrieval tasks with 128 bits under different numbers of class prototypes $K$.

## 5. Conclusion

In this work, we reveal and study a practical but less-touched problem in cross-modal hashing (CMH), i.e., streaming-media hashing retrieval. To overcome this issue, we propose a novel CMH paradigm dubbed SHE, which could train continuously incoming streaming-media data in parallel. The proposed SHE mainly consists of three core modules, i.e., Knowledge Library Mining module (KLM), Knowledge Library Transfer module (KLT), and Discriminative Hashing Learning module (DHL). Specifically, KLM extracts essential class prototypes, thereby constructing an implicit knowledge library for each modality. KLT aligns the knowledge of the newly arrived modality with the historical knowledge library, thereby preserving semantic consistency across modalities. DHL maximizes intra-class semantic relevance and inter-class semantic disparity, resulting in compact and discriminative hash codes. Extensive experiments on four benchmark datasets highlight the superiority of our SHE compared to 14 state-of-the-art methods.

## Acknowledgements

This work is supported in part by the National Natural Science Foundation of China under Grant No. 62372315, the Sichuan Science and Technology Planning Project under Grants No. 2024NSFTD0049, 2024ZDZX0004, 2024YFHZ0089, the Chengdu Science and Technology Project (Grant no. 2023-XT00-00004-GX), and the Sichuan Science and Technology Miaozi Program (Grant no. MZGC20240057).

## Impact Statement

Most existing cross-modal retrieval (CMR) methods commonly assume synchronous availability of media data, ignoring the presence of streaming-media data (SMD), i.e., continuously incoming data. Although some real-valued methods independently train each modality to improve flexibility in handling SMD, their low retrieval efficiency falls short of large-scale data retrieval. This paper proposes a novel hashing paradigm to achieve asynchronous retrieval for SMD, offering a more practical and portable solution for real-world applications.

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

# APPENDIX

This document provides supplementary material to our manuscript, which aims to offer a comprehensive understanding of our SHE framework. Specifically, we mainly present the dataset description, additional experimental results, more detailed analysis about the media learning sequence, the analysis about the impact of bit length, and the discussion of limitations of this work to support the main submission. For convenience, we provided *jumpable links* to each supplementary section in Tab.6.

*Table 6.* The hyperlinks for each supplementary section.

## A. Dataset

The following is a detailed introduction to each dataset: **Wikipedia** is a dataset comprising 2,866 image-text pairs with 10 categories. For our experiment, we randomly allocate 2,173 and 693 pairs for training and testing, respectively, while all pairs are used as the retrieval database. **NUS-WIDE** is a dataset comprising over 260,000 image-text pairs with 81 categories. In this work, we utilize a subset of 10,000 pairs with 10 semantic labels to perform experiments. In more detail, we randomly assign 8,000 and 2,000 pairs for training and testing, respectively, while all pairs are used as the retrieval database. **XMedia** is a dataset covering five modalities with 20 categories, comprising 5,000 image-text pairs, 1,000 audio clips, 500 3D models, and 1,143 videos. For our experiment, we select 4,000 image-text pairs, 800 audio clips, 400 3D models, and 969 videos as the training set, while the remaining samples are used as the test set. Besides, all samples from the dataset are included in the retrieval database. **XMediaNet** is a dataset covering five modalities with 20 categories, comprising 40,000 image-text pairs, 10,000 audio clips, 2,000 3D models, and 10,000 videos. For our experiment, we select 32,000 image-text pairs, 8,000 audio clips, 1,600 3D models, and 8,000 videos as the training set, while the remaining samples are used as the test set. Besides, all samples from the dataset are included in the retrieval database.

For Wikipedia and NUS-WIDE, the initial features of the image modality are extracted using the pre-trained VGG-19 (Simonyan, 2014) model, while the features of the text modality are obtained using the pre-trained Doc2vec (Lau & Baldwin, 2016) model. For XMedia and XMediaNet, their original features are provided by the authors (Wang & Peng,

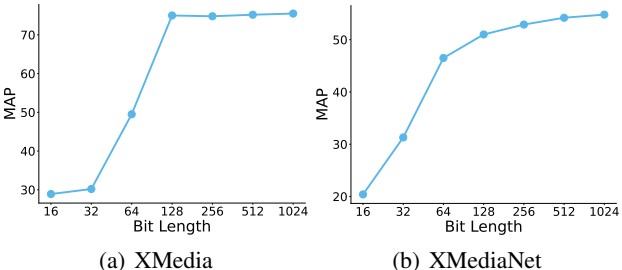

(a) XMedia      (b) XMediaNet

*Figure 6.* The MAP scores with different bit lengths on the XMedia and XMediaNet datasets.

2021).

## B. The Impact of Bit Length

To examine the impact of bit length on retrieval performance, we perform experiments on the XMedia and XMediaNet datasets. The corresponding mean MAP scores of all cross-modal retrieval tasks are illustrated in Fig.6. The results reveal the following insights: 1) Hash codes with longer bit lengths can carry more semantic information, thus enjoying better performance. 2) Given that the XMedia and XMediaNet datasets contain five modalities with considerable differences, hash codes with shorter bit lengths (e.g., 16 and 32 bits) may fail to effectively capture the underlying semantic features, leading to subpar performance.

## C. Precision-recall Curves

To comprehensively evaluate the retrieval performance of our proposed SHE, we provide precision-recall curves with different bit lengths (i.e., 16, 32, and 64 bits) on the Wikipedia and NUS-WIDE datasets. As shown in Fig.7, SHE consistently exhibits higher precision at almost all recall rates compared to SOTA methods, exhibiting its stronger discriminative power and retrieval performance.

## D. The Impact of Media Learning Sequence

In the manuscript, we have delved into the impact of media learning sequences with 128 bits on the XMedia and XMediaNet datasets in terms of mean MAP scores of all cross-modal retrieval tasks. To offer a more detailed analysis, we provide the MAP scores for each cross-modal retrieval task in Tab.7. From the results, we can get the same findings as the manuscript. Besides, we also find that when the audio modality is selected as the first available modality on the XMedia dataset, the performance on the audio, 3D, and video modalities degrades significantly. This may be due to: 1) The audio modality being represented by low-dimensional (29D) MFCC features with weak expressive power, leading to poor-quality knowledge initialization. 2)

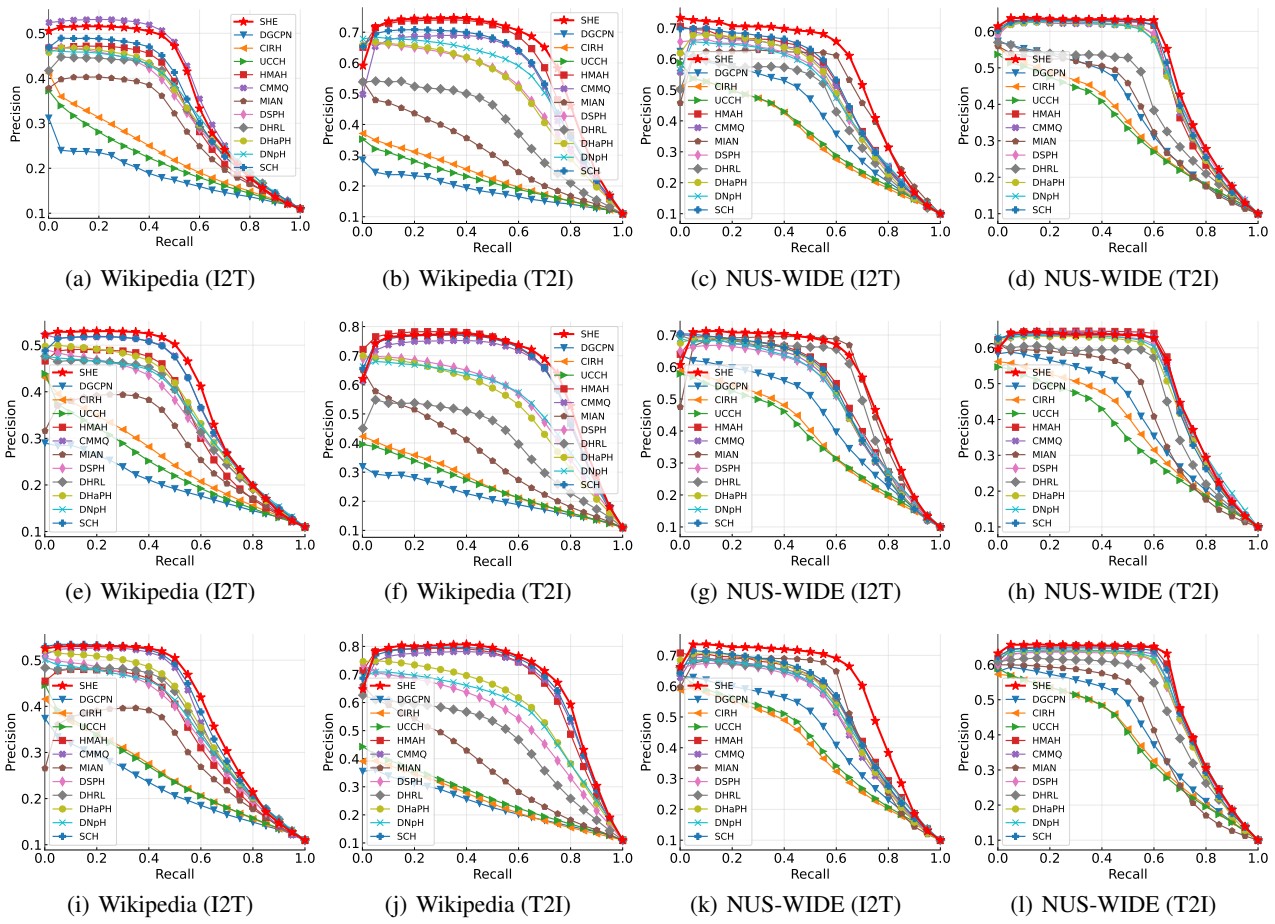

*Figure 7.* Precision-recall curves with different bit lengths on the Wikipedia and NUS-WIDE datasets, where (a-d), (e-h), and (i-l) show the curve results under 16 bits, 32 bits, and 64 bits, respectively.

*Table 7.* The MAP scores with 128 bits for different choices of the first available modality on the XMedia and XMediaNet datasets.

| Dataset | | Image | | | | Text | | | | Audio | | | | 3D | | | | Video | | | | Avg |
|---|---|---|---|---|---|---|---|---|---|---|---|---|---|---|---|---|---|---|---|---|---|---|
| | | Text | Audio | 3D | Video | Image | Audio | 3D | Video | Image | Text | 3D | Video | Image | Text | Audio | Video | Image | Text | Audio | 3D | |
| XMedia | Image | 92.1 | 87.3 | 89.6 | 87.2 | 95.8 | 92.0 | 94.0 | 91.7 | 65.3 | 66.5 | 64.8 | 62.3 | 70.2 | 70.4 | 67.7 | 66.4 | 60.7 | 61.3 | 57.2 | 58.4 | 75.0 |
| | Text | 91.8 | 86.7 | 88.4 | 87.4 | 95.4 | 91.3 | 93.6 | 92.5 | 63.4 | 64.5 | 62.3 | 60.6 | 68.6 | 69.4 | 64.0 | 64.7 | 60.5 | 62.2 | 56.5 | 58.2 | 74.1 |
| | Audio | 90.0 | 6.0 | 8.4 | 61.1 | 91.4 | 5.5 | 7.8 | 56.5 | 10.4 | 10.3 | 7.0 | 11.5 | 9.6 | 13.7 | 5.8 | 10.7 | 40.7 | 39.1 | 5.3 | 6.7 | 24.9 |
| | 3D | 91.6 | 71.9 | 84.1 | 84.0 | 94.6 | 75.0 | 88.1 | 86.1 | 52.9 | 55.1 | 49.8 | 47.5 | 59.7 | 58.6 | 51.2 | 54.5 | 59.8 | 60.5 | 47.2 | 54.4 | 66.3 |
| | Video | 91.9 | 85.0 | 88.9 | 87.4 | 95.6 | 89.9 | 93.9 | 92.0 | 63.0 | 63.2 | 60.8 | 59.3 | 68.3 | 69.3 | 63.2 | 64.3 | 60.8 | 61.6 | 55.6 | 57.2 | 73.6 |
| XMediaNet | Image | 75.1 | 70.6 | 70.3 | 72.8 | 70.5 | 62.8 | 63.5 | 65.8 | 42.4 | 39.8 | 36.7 | 38.2 | 41.2 | 39.3 | 34.0 | 37.7 | 43.1 | 41.0 | 36.4 | 38.3 | 51.0 |
| | Text | 73.4 | 67.7 | 68.1 | 71.4 | 69.0 | 60.8 | 61.2 | 64.0 | 41.1 | 38.4 | 35.4 | 36.9 | 38.2 | 36.6 | 32.8 | 35.3 | 43.3 | 41.0 | 36.9 | 37.9 | 49.5 |
| | Audio | 74.9 | 63.8 | 67.8 | 72.6 | 69.9 | 55.4 | 60.2 | 65.0 | 42.4 | 39.5 | 35.7 | 38.2 | 38.9 | 36.8 | 27.6 | 35.8 | 43.1 | 41.5 | 33.1 | 37.2 | 49.0 |
| | 3D | 74.6 | 70.0 | 67.4 | 72.3 | 69.9 | 62.9 | 60.2 | 65.0 | 41.1 | 39.0 | 33.6 | 37.3 | 36.9 | 35.2 | 30.9 | 34.5 | 42.7 | 41.1 | 37.3 | 36.2 | 49.4 |
| | Video | 75.4 | 70.4 | 70.4 | 73.3 | 71.0 | 62.8 | 63.5 | 66.0 | 41.1 | 38.8 | 35.8 | 36.9 | 40.4 | 38.7 | 34.1 | 37.6 | 43.6 | 41.1 | 36.7 | 39.0 | 50.8 |

The insufficient number of training samples in the audio, 3D, and video modalities, which causes underfitting.

# E. Limitations

This paper proposes a novel cross-modal hashing (CMH) retrieval framework tailored for streaming-media hashing retrieval. To the best of our knowledge, this could be the first work to address the problem of streaming-media data (SMD) using a CMH paradigm. However, we have not yet accounted for the scenario that SMD may contain noisy annotations, which may be one of our future research directions.

