# OpenReview forum: "SHE: Streaming-media Hashing Retrieval"
_ICML.cc/2025/Conference — ICML 2025 poster_

### Official Review · Reviewer_LZBc · 2025-03-07

**Overall Recommendation:** 4

**Summary:**

Existing CMH methods often implicitly assume that all modalities are prepared before processing. However, in practice applications (such as multi-modal medical diagnosis), it is very challenging to collect paired multi-modal data simultaneously. Specifically, they are collected chronologically, forming streaming-media data (SMA). To handle this, all previous CMH methods require retraining on data from all modalities, which inevitably limits the scalability and flexibility of the model. For this issue, this paper proposes a novel CMH paradigm named Streaming-media Hashing rEtrieval (SHE) that enables parallel training of each modality. Specifically, SHE proposes a knowledge library mining module (KLM) that extracts a prototype knowledge library for each modality, thereby revealing the common distribution of each modality. Then, SHE proposes a knowledge library transfer module (KLT) that uses the historical knowledge library to update and align new knowledge to ensure semantic consistency. Finally, to enhance the intra-class semantic relevance and inter-class semantic differences, SHE develops a discriminative hash learning module (DHL). In general, this paper is significantly innovative and has important value in practical applications.

**Claims And Evidence:**

Yes, the claim that SHE can train all modalities in parallel and improve the flexibility and scalability of cross-modal retrieval is supported by extensive experimental evidence on four datasets. The improvement in retrieval performance is shown in experimental results.

**Essential References Not Discussed:**

Key references are cited in this paper, but the authors are recommended to cite and discuss more cross-modal retrieval methods that specifically handle streaming-media data to enrich the research context of the current work.

**Experimental Designs Or Analyses:**

Yes, the experimental setting is reasonable. The dataset containing five modalities is used for the experiment and compared with other state-of-the-art (SOTA) methods to fully demonstrate the superiority of SHE under streaming-media data. Additionally, The ablation experiment demonstrates the effectiveness of each component of SHE.

**Methods And Evaluation Criteria:**

Yes, the proposed SHE is suitable for solving the cross-modal retrieval problem under streaming-media data. The selected baseline datasets and evaluation metrics such as the Mean Average Precision (MAP) score are suitable for evaluating the superiority of the proposed SHE. This paper adopts a reasonable experimental setting and compares it with the state-of-the-art methods, which is helpful to verify the effectiveness of SHE.

**Other Comments Or Suggestions:**

1. The authors are recommended to provide more detailed explanations of key loss terms to enhance readability and help readers better grasp the design motivations.
2. The authors are recommended to provide more discussion and comparison with more cross-modal retrieval methods that specifically handle streaming-media data.

**Other Strengths And Weaknesses:**

Strengths:

1. This paper introduces a Streaming-media Hash rEtrieval (SHE) framework specifically designed for streaming-media applications, offering a unique and innovative research perspective while demonstrating its practical value in the field of cross-modal retrieval.
2. The experimental results show that SHE achieves superior retrieval performance across multiple benchmark datasets. Additionally, ablation studies confirm the contributions of each module to the overall model performance.
3. This paper demonstrates a high degree of clarity in its writing style and organizational structure, with well-articulated and explicit experimental motivations.

Weaknesses:
1. Some loss function formulations are relatively complex, which may hinder the understanding of the core optimization objectives. The authors should provide more detailed explanations of key loss terms to enhance readability.
2. The authors do not explain why only the nearest prototype is considered when mining the knowledge library.

**Questions For Authors:**

1. This paper mines a knowledge library from each modality by KLM, and then aligns the knowledge libraries of different modalities through the KLT module to achieve semantic alignment. Can the author explain the benefits of doing so?
2. When mining the knowledge library, this paper only considers the nearest prototype with the same category. What are the benefits of doing so and can it be compared with other methods?
3. Apart from retrieval efficiency, could the authors provide other advantages of SHE compared to MARS?

**Relation To Broader Scientific Literature:**

This paper positions the proposed Streaming-Media Hashing rEtrieval (SHE) within the field of cross-modal retrieval (CMR), comparing SHE with numerous cross-modal hashing (CMH) methods and three real-valued CMR methods. The study underscores the limitations of existing CMH methods in handling streaming-media data and introduces SHE as a novel hashing framework to address these challenges.

**Theoretical Claims:**

Yes, the theoretical claims that SHE mines a knowledge library for each modality as a medium for maintaining semantic consistency is feasible. This is because the knowledge library can capture the common distribution information in streaming-media data, and the knowledge libraries of different modalities are aligned through the Knowledge Library Transfer (KLT) module, and using the knowledge library as a medium can ensure the consistency of cross-modal semantics.

---

> ### Author Rebuttal · Authors · 2025-04-01
>
> Thanks for your valuable comments. Our responses are listed below.
>
> **R-Weakness-1**: To enhance readability and understanding of our proposed SHE, we provide a detailed explanation of some key losses, which can be found at **`R-Weakness-3’** in [Reviewer TSiu](https://openreview.net/forum?id=JqLKV0L5hM&noteId=KyOMFKYOSw).
>
> **R-Weakness-2**: When mining the knowledge library, if all class prototypes are considered simultaneously, a sample may be influenced by multiple prototypes, leading to an unstable optimization objective. Thus, we only consider the nearest prototype. Besides, to verify the effectiveness of the above behavior, we conduct experiments on the XMedia and XMediaNet datasets with 128 bits by comparing the consideration of only the nearest prototype versus considering all prototypes. Specifically, we replace the KLM loss with the KLM* loss when mining the knowledge library, and the KLM* loss can be formulated as:
>
> $L_{klm^*} = -\frac{1}{n_m} \sum_{i=1}^{n_m} \log \left( \frac{1}{K} \sum_{k=1}^{K}  s\left(x_m^i | k\right) \right).$
>
> The results are shown as follows:
>
> |Loss|XMedia|XMediaNet|
> |:-:|:-:|:-:|
> |$L_{klm*}$|73.3|46.9|
> |$L_{klm} $|75.0|51.0|
>
> The results show that only considering the nearest prototype with the same category when mining the knowledge library can enjoy a clear optimization objective and a better performance.
>
> **R-Suggestion-1**: We provide detailed explanations of key loss terms in **`R-Weakness-3’** of [Reviewer TSiu](https://openreview.net/forum?id=JqLKV0L5hM&noteId=KyOMFKYOSw).
>
> **R-Suggestion-2**: We investigate the latest studies and found that there are no other CMH works to handle cross-modal retrieval (CMR) in streaming-media scenarios (SMC). Here, we discuss some real-valued representation-based CMR methods, including SDML[1] and DRCL[2], which are specifically designed to accomplish CMR in SMC. 1) These methods are real-valued based representations, which involve high computational complexity and memory overhead, making it difficult to meet the fast retrieval demands of large-scale datasets. 2) They deploy either a randomly initialized common space or an identical transformation weight matrix to guide semantic alignment, which overlooks the discrepancies among modalities. Since the randomly initialized common space or learned transformation weight matrix may retain modality-specific features that are not very relevant to the new modality, directly applying it may lead to semantic discrepancies. By contrast, our SHE transfers the baseline knowledge library to the new modality, thereby obtaining more semantically consistent representations.
>
> **R-Question-1**: The KLM module can mine a knowledge library to preserve the semantic information of streaming data, which can serve as a medium to maintain semantic consistency while reducing the computational burden of maintaining historical data. The KLT module can transfer semantic information extracted from new modalities to the benchmark knowledge library. This allows our SHE to process new media without retraining the entire historical media data, thereby reducing computational complexity while ensuring semantic consistency between streaming modal data.
>
> **R-Question-2**: We provide the explanation of why only considering the nearest prototype when mining the knowledge library, and perform a comparison with other methods. Please see **`R-Weakness-2’** for details.
>
> **R-Question-3**: To improve the efficiency of handling streaming-media data, MARS leverages the shared label parsing module to achieve semantic alignment without interaction across modalities. However, the label parsing module may retain modality-specific features that are less relevant to the new modality, and directly using it may lead to semantic bias. By contrast, our SHE proposes a KLT module to transfer the baseline knowledge library to the new modality, thereby obtaining more semantically consistent representations. To further demonstrate the advantages of SHE, we integrate the idea of using a knowledge library as a medium to maintain semantic consistency into MARS, and found it beneficial to do so. For detailed experiments and analysis, see **`R-Question-2’** in [Reviewer UdWL](https://openreview.net/forum?id=JqLKV0L5hM&noteId=G4Nh3s9sQe).
>
> **Reference**
>
> [1] Hu P, Zhen L, Peng D, et al. Scalable deep multimodal learning for cross-modal retrieval[C]//Proceedings of the 42nd international ACM SIGIR conference on research and development in information retrieval. 2019: 635-644.
>
> [2] Pu R, Qin Y, Peng D, et al. Deep Reversible Consistency Learning for Cross-modal Retrieval[J]. IEEE Transactions on Multimedia, 2025.

---

### Official Review · Reviewer_UdWL · 2025-03-10

**Overall Recommendation:** 4

**Summary:**

This paper proposes a novel CMH paradigm, specifically designed for cross-modal retrieval in streaming-media scenarios. The proposed SHE framework comprises three key modules: the Knowledge Library Mining (KLM) module, the Knowledge Library Transfer (KLT) module, and the Discriminative Hash Learning (DHL) module. Specifically, KLM constructs a knowledge library for every modality by extracting essential prototype knowledge, thereby serving as a semantic consistency maintenance medium. KLT aims to adaptively align the knowledge library extracted from the newly arrived modality with the historical knowledge library, thereby capturing the semantic consistency of multiple modalities. DHL enhances the quality of hash codes by maximizing intra-class semantic relevance while amplifying inter-class semantic disparity. Extensive experimental evaluations demonstrate that the proposed method effectively handles cross-modal retrieval in streaming-media scenarios, offering a flexible and scalable solution for real-world applications.

**Claims And Evidence:**

SHE effectively handles cross-modal retrieval in streaming-media scenarios, which is supported by extensive experimental evaluations on four benchmark datasets. Besides, SHE can train each modality in parallel, which is supported by the reasonable module design. Although each modality is learned in parallel, semantic alignment can be achieved by using knowledge libraries as the bridge.

**Essential References Not Discussed:**

Further discussion on cross-modal hashing methods, particularly techniques addressing cross-modal retrieval in streaming-media scenarios, would be beneficial.

**Experimental Designs Or Analyses:**

The experimental designs are sound, with suitable benchmark datasets, appropriate evaluation metrics, well-defined experimental objectives, and reasonable experimental analysis. Extensive experiments and analyses are conducted to comprehensively evaluate assess the effectiveness of SHE, such as investigations into the impact of media learning sequences, similarity boundaries, and the scale of the knowledge library.

**Methods And Evaluation Criteria:**

The proposed SHE framework effectively addresses the challenges of cross-modal retrieval, particularly in streaming-media scenarios. The selected benchmark datasets are well-suited for evaluating the effectiveness of SHE, as two of them contains five modalities and are unpaired, which accurately simulate the characteristics of streaming-media data. Additionally, the evaluation metrics employed in this study are widely recognized in the field of cross-modal retrieval and are highly appropriate for assessing the performance of SHE and the baseline methods.

**Other Comments Or Suggestions:**

1)	Further discussion on cross-modal hashing methods, particularly techniques addressing cross-modal retrieval in streaming-media scenarios, would be beneficial.

2)	Some sections could be written and structured more clearly.

**Other Strengths And Weaknesses:**

Strengths:

1)	Sound in originality. This paper proposes a novel CMH framework named SHE specifically designed for streaming-media data, which is a highly valuable application scenario in cross-modal retrieval.

2)	Reasonable technical route. To enable parallel training of different modalities while maintaining semantic consistency, SHE employs the KLM module to construct a dedicated knowledge library for each modality, preserving semantic information from streaming-media. Subsequently, the KLT module aligns the newly extracted knowledge library with the baseline knowledge library, ensuring cross-modal semantic coherence.

3)	Reasonable and comprehensive experimental arrangements. Extensive experiments are conducted on four datasets to verify the effectiveness and superiority of this method.

Weaknesses:

1)	This paper designates the knowledge library extracted from the first modality as the benchmark which may introduce a degree of arbitrariness. In other words, the study has not yet evaluated the quality of the knowledge libraries extracted from different modalities, which could impact the overall effectiveness of the proposed method.

2)	Some sections could benefit from clearer writing and organization.

3)	From Fig. 3, the media learning sequence has a significant impact on performance, but no possible solution is provided.

**Questions For Authors:**

1)	This paper designates the knowledge library extracted from the first modality as the benchmark which may introduce a degree of arbitrariness. This paper would benefit from comparing and evaluating the quality of knowledge libraries.

2)	Would integrating the proposed idea of using a knowledge library as a medium to maintain semantic consistency into existing methods be beneficial?

3)	Except the regularization approach applied to the knowledge library in this paper, could the authors provide a comparative analysis with other existing regularization techniques?

**Relation To Broader Scientific Literature:**

This paper proposes a novel Cross-Modal Hashing (CMH) framework, termed Streaming-media Hashing rEtrieval (SHE), which is highly relevant to the domain of cross-modal retrieval. To be specific, this paper aims to analyze the limitations and shortcomings of existing research in streaming-media scenarios and propose corresponding solutions, thereby highlighting the advantages of SHE.

**Theoretical Claims:**

SHE can handle unpaired multimodal data, which is well-grounded. SHE adopts a learning framework without modality interaction and utilizes class prototypes as an abstract knowledge library to preserve the semantic information of streaming-media data. This enables effective semantic representation learning even for unpaired multimodal data.

---

> ### Author Rebuttal · Authors · 2025-04-01
>
> Thanks for your constructive review. Our responses are listed below.
>
> **R-Weakness-1**: To evaluate the quality of knowledge mined from different modalities, we design a score, which can be found at **`R-Questions’** in [Reviewer cJMB](https://openreview.net/forum?id=JqLKV0L5hM&noteId=i9vSJNoRoW).
>
> **R-Weakness-2**: To make readers understand the work more clearly, we reorganize the writing, such as the design of losses. Specifically, we explain Knowledge Library Regularization (KLR) loss and Knowledge Library Transfer (KLT) loss in detail, which can be found at **`R-Weakness-3’** in [Reviewer TSiu](https://openreview.net/forum?id=JqLKV0L5hM&noteId=KyOMFKYOSw).
>
> **R-Weakness-3**: It is a constructive suggestion. To solve this, we first design a score to evaluate the quality of knowledge mined from different modalities, which can be found in **`R-Questions’** in [Reviewer cJMB](https://openreview.net/forum?id=JqLKV0L5hM&noteId=i9vSJNoRoW). Then, due to the word limit, we provide a simple solution here. If we observe that the score of the current modality is too low, we can wait for subsequent modalities to arrive. Once a modality with sufficient quality is detected, it can be selected as the initial modality. According to your suggestion, we will explore how to alleviate negative effects caused by low-quality first modalities in the future.
>
> **R-Suggestion-1**: We investigate the latest studies and found that there are no other CMH works to handle cross-modal retrieval (CMR) in streaming-media scenarios (SMC). Here, we discuss some real-valued representation-based CMR methods, including SDML[1] and DRCL[2], which are specifically designed to accomplish CMR in SMC. 1) These methods are real-valued based representations, which involve high computational complexity and memory overhead, making it difficult to meet the fast retrieval demands of large-scale datasets. 2) They deploy either a randomly initialized common space or an identical transformation weight matrix to guide semantic alignment, which overlooks the discrepancies among modalities. Since the randomly initialized common space or learned transformation weight matrix may retain modality-specific features that are not very relevant to the new modality, directly applying it may lead to semantic discrepancies. By contrast, our SHE transfers the baseline knowledge library to the new modality, thereby obtaining more semantically consistent representations.
>
> **R-Suggestion-2**: We reorganize the writing so that readers can understand the work more clearly, such as the design of losses. The details can be found at **`R-Weakness-3’** in [Reviewer TSiu](https://openreview.net/forum?id=JqLKV0L5hM&noteId=KyOMFKYOSw).
>
> **R-Question-1**: We have designed a score to evaluate the quality of knowledge libraries. Please see **`R-Questions’** in [Reviewer cJMB](https://openreview.net/forum?id=JqLKV0L5hM&noteId=i9vSJNoRoW).
>
> **R-Question-2**: As you suggested, we conduct experiments on the XMedia and XMediaNet datasets and found that it is effective to incorporate the above idea into existing methods (i.e., MARS). Specifically, we replace the original label parsing module in MARS with the knowledge library to guide representation learning and add the Knowledge Library Transfer (KLT) loss to maintain semantic consistency. The new objective of MARS can be formulated as:
>
>  $L=L_{ori}+L_{klt},$
>
>  where $L_{ori}$ is the original objective of MARS and $L _{klt}$ is the KLT loss. The results are shown as follows:
>
> |Objective|XMedia|XMediaNet|
> |:-:|:-:|:-:|
> |$L _{ori}$|72.0|46.1|
> |$L$|73.8|51.7|
>
> The results can show that using knowledge libraries as a medium to maintain semantic consistency can be integrated into existing methods.
>
> **R-Question-3**: In fact, the KLR loss is not our main innovation, so we only provide a simple comparative Analysis. As you suggested, we provide another regularization technique, i.e., orthogonal regularization (OR). Specifically, OR aims to impose the constraint that inter-class prototypes are mutually orthogonal, which can be formulated as:
>
> $L_{or} =   \frac{1}{CK} \sum_{c=1}^{C} \sum_{k=1}^{K}\sum_{c'=1,c'\ne c}^{C} \sum_{k'=1}^{K} \left | \Gamma (p^{c,k}_m,p^{c',k'}_m) \right |,$
>
> where $\Gamma$ is the cosine similarity function. We replace Knowledge Library Regularization (KLR) loss with OR loss and conduct experiments on the XMedia and XMediaNet datasets with 128 bits, and the results are shown as follows:
>
> |Regularization|XMedia|XMediaNet|
> |:-:|:-:|:-:|
> |No|73.8|33.3|
> |OR|73.9|40.0|
> |KLR|75.0|51.0|
>
> The results show that OR is effective but inferior to the regularization technique used in our paper. In the future, we will further explore better regularization techniques to contribute to our proposed SHE.
>
> **Reference**
>
> [1] Scalable deep multimodal learning for cross-modal retrieval, SIGIR. 2019: 635-644.
>
> [2] Deep Reversible Consistency Learning for Cross-modal Retrieval, TMM, 2025.

---

### Official Review · Reviewer_cJMB · 2025-03-11

**Overall Recommendation:** 4

**Summary:**

This work addresses a less studied but practically valuable problem in cross-modal retrieval, i.e., streaming-media hashing retrieval. The paper points out the key challenge is to preserving cross-modal interactions. Through knowledge library mining on existing modalities and knowledge transferring only on the subsequently arrived modalities, the proposed training framework requires no re-training on the historical data but successfully establishes cross-modal knowledge alignment, hence reducing training complexity. Experiments on various datasets have validated the effectiveness of the proposed method.

## update after rebuttal
The rebuttal from authors solved my concerns regarding clarity. Therefore, I keep my score as 4 towards acceptance.

**Claims And Evidence:**

Yes, claims regarding the effectiveness of each part of the paper, including the three modules and all the loss objectives are well-supported by clear experimental evidences, including benchmarks with various number of modalities and comprehensive ablation study on all evaluated datasets.

**Essential References Not Discussed:**

Key references are well-discussed.

**Experimental Designs Or Analyses:**

The paper's method is verified comprehensively and analyzed thoroughly to establish its validity.

**Methods And Evaluation Criteria:**

The proposed method is established with solid analysis on the characteristics of streaming-media data, while the used metrics (mAP, P-R Curves) are widely-used evaluation protocols in cross-modal hashing that can also effectively evaluate the studied problem. The selected datasets are widely-used for cross-modal hashing with multiple modalities, hence suitable for the evaluation of the proposed method.

**Other Comments Or Suggestions:**

1. Typos: a) Line16: “in practice applications”. b) Line 39 (the right column) “such as such as”.
2. It's better to add some explanations about the Eq. 8 loss objective and the similarity boundary.

**Other Strengths And Weaknesses:**

Strengths:
1. The paper addresses the novel and critical problem of streaming-media hashing retrieval with clear analysis of its potential challenges.
2. The method is clearly motivated and designed based on the characteristics of streaming-media data.
3. Experiments compare with state-of-the-art methods and display significant advantages for retrieval with increasing modalities. Ablation study and other analyses further emphasize the validity and effectiveness of the method. Results with different media sequence verifies the general effectiveness with different initial modalities.

Weaknesses:
1. The authors may include descriptions of the training procedure for the compared methods.

**Questions For Authors:**

The model’s performance can vary upon the choice of the initial modality, which is sometimes infeasible to be simply replaced by another in real applications. What kind of improvements are significant to alleviate such effects caused by low-quality first modalities?

**Relation To Broader Scientific Literature:**

The work can be applied to cross-modal hashing with increasing data. Particularly, this paper contributes to the problem of increasing modalities and can also potentially improve the similarity learning of current multi-modal hashing methods.

**Theoretical Claims:**

The paper does not contain proofs/theoretical claims.

---

> ### Author Rebuttal · Authors · 2025-04-01
>
> We appreciate your valuable feedback. Our responses are listed below.
>
> **R-Weaknesses**: To make the experiment more rigorous, a detailed description of the training procedure for the compared methods is provided. Specifically, there are two cases in the training. 1) For the Wikipedia and NUS-WIDE datasets, all baselines directly conduct training without any preprocessing or modification, as the data in these datasets consists of image-text pairs. 2) For the XMedia and XMediaNet datasets, all baselines except for MARS adopt label-based repeat sampling to construct pseudo-instance pairs, and then conduct training on every two modalities, as the data in these datasets consists of independent instances (unpaired and inconsistent in number). MARS directly conducts training without any preprocessing or modification, as it can process each modality independently.
>
> **R-Suggestion-1**: We carefully check our paper and ensure that similar mistakes are not made.
>
> **R-Suggestion-2**: To enhance the understanding of the loss of Eq. 8 (i.e., Knowledge Library Transfer loss, KLT), we add some explanations to further elaborate on it. The KLT loss is formulated as:
>
> $L_{klt} = -\frac{1}{CK} \sum_{c=1}^{C} \sum_{k=1}^{K}
> \log \left [ min(1,max(0,\Gamma(p_1^{c,k},p_m^{c,k}) - \sigma +1)) \right ].$
>
> KLT aims to transfer the semantic information in the newly mined knowledge to the benchmark knowledge library, thereby achieving cross-modal semantic consistency. As is shown in the above equation, we encourage the prototypes mined from the newly coming modality to align with the prototypes with the same semantics in the benchmark knowledge library by maximizing their similarity. Notably, we focus on maintaining a certain level of semantic similarity between them by setting a similarity boundary rather than forcing them to be identical. That is, the above learning objective is $\Gamma(p_1^{c,k},p_m^{c,k}) \ge \sigma$, which relaxes the alignment requirement and can further improve the representation ability of multimodal data.
>
>
> **R-Questions**: It is a constructive suggestion. To alleviate negative effects caused by low-quality first modalities, we first design a score to evaluate the quality of knowledge mined from different modalities. For the $m$-th modality, the score could be formulated as follows:
>
> $S_{m}=\frac{1}{2n_m} \sum_{i=1}^{n_m}  \max_{k=1, \ldots, K} s\left(x_m^i | k\right) +\frac{1}{2CK} \sum_{c=1}^{C} \sum_{k=1}^{K} \frac{ \sum_{k'=1,k' \ne k }^{K}e^ {\Gamma(p^{c,k}_m,p^{c, k'}_m)}} {\sum _{c'=1}^{C}\sum _{k'=1}^{C}{e^ {\Gamma(p^{c,k}_m,p^{c',k'}_m)}}-e},$
>
> where $s(x_m^i | k)=\frac{\sum_{c=1}^{C} y_m^{i,c}\cdot \Gamma(b_m^i,p^{c,k}_m)+1}{2}$ and $\Gamma(\cdot,\cdot)$ is the cosine similarity function. In the above equation, the first item is responsible for reflecting the consistency between the mined knowledge and sample representations, while the second item is responsible for reflecting the distinction among inter-class knowledge. The higher the score is, the more the mined knowledge reflects the semantic information in the samples and the more discriminable it is, that is, the higher quality the modality owns. To verify the effectiveness of the proposed score, we conduct experiments on the XMedia dataset with 128 bits. The results are shown as follows:
>
> |Indicator|Image|Text|Audio|3D|video|
> |:-:|:-:|:-:|:-:|:-:|:-:|
> |Score|0.763|0.763|0.560|0.747|0.762|
> |MAP|75.0|74.1|24.9|66.3|73.6|
>
> From the results, it can be observed that the proposed score can accurately reflect the quality of the modalities. Due to the word limit, a simple strategy is provided here. In real applications, if we observe that the score of the current modality is too low, we can wait for subsequent modalities to arrive. Once a modality with sufficient quality is detected, it can be selected as the initial modality. According to your suggestion, we will explore how to alleviate negative effects caused by low-quality first modalities in the future.

---

### Official Review · Reviewer_TSiu · 2025-03-11

**Overall Recommendation:** 3

**Summary:**

While most previous research in CHM have assumed that all modalities are prepared before processing, the authors propose a novel CHM paradigm named Streaming-media Hashing rEtrieval (SHE) that enables parallel training of each modality for streaming-media data, where the data is collected chronologically.
The paradigm includes a knowledge library mining module and a knowledge library transfer module to jointly extract an implicit knowledge library from the new incoming data and align the commonality distribution from the new knowledge with ones from the historical knowledge library and also a discriminative hashing learning module to enhance intra-class semantic relevance and inter-class semantic disparity.
This paradigm is tested over 4 common benchmark datasets streaming-media retrieval, and the results show its superiority over 14 state-of-the-art methods.

**Claims And Evidence:**

Yes, the claims in the submission are well supported by clear and convincing evidence.

**Essential References Not Discussed:**

It is suggested to list the references about the previous CMH methods in introduction.

**Experimental Designs Or Analyses:**

Yes, the experimental designs and analyses are valid.

**Methods And Evaluation Criteria:**

Yes, the proposed methods and evaluation criteria make sense for the problem or application at hand.

**Other Comments Or Suggestions:**

Typos
- Line 39 right: two ‘such as’s
- Line 99 right: ‘knowledge library mining module (KLM) module’ module is included in the abbreviation
- Line 205 right: ‘where $p_m^c$ denotes K the normalized and binary prototype for the c-th class’ unnecessary ‘K’

**Other Strengths And Weaknesses:**

Strengths
- The paper is well-constructed and easy to read.
- The experiments are conducted over 4 experiments and 14 state-of-the-art methods.

Weaknesses
- In section 3.2, the authors use the notation KLR loss before explaining what it is. It’s better to address it with its full name and notify the readers this will be explained later.
- In introduction, authors describe many previous CMH methods with all sorts of shortcomings but fail to list the actual methods. Also, the authors only point out emergency medical aid for practical application scenarios, which seems insufficient.
- The loss function formulation in this paper lacks detailed explanation. It’s better to explain the loss function design more specifically.
- The authors mention that the proposed method enables parallel training of each modality, making training more efficient and require less storage. But this is not shown in the experiments.

**Questions For Authors:**

please refer to the weakness part.

**Relation To Broader Scientific Literature:**

The proposed SHE aims to solve the problem of training streaming-media data, which isn’t prepared before processing. SHE introduces an evolving knowledge library to avoid training redundancy and also improve training efficiency.

**Theoretical Claims:**

Yes.

---

> ### Author Rebuttal · Authors · 2025-04-01
>
> Thanks for your valuable comments. Our responses are listed below.
>
> **R-Weakness-1**: We have revised this issue, providing the relevant explanation when KLR first appears.
>
> **R-Weakness-2**: In the introduction section, we list CMH methods (Such as UCCH, DHaPH, DCH-SCR[1], SCH, and CICH[2]) to illustrate the shortcomings of existing CMH research. Here, we briefly discuss some recent CMH works below: 1) UCCH aims to implicitly capture the semantic relevance between modalities by mining the co-occurrence relationship between multimodal data (such as image-text pairs) and their initial distribution characteristics. 2) DCH-SCR[1] deploys a ranking alignment mechanism to capture the close semantic relationship between modalities, which preserves the semantic similarity between tags and feature levels. 3) CICH[2] designs a prototypical semantic similarity coordination module to globally rebuild partially-observed cross-modal similarities under an asymmetric learning scheme. Notably, these CMH methods implicitly assume that all modalities are prepared before processing and adopt joint learning to achieve cross-modal semantic alignment. In practical application scenarios, it is challenging to collect data of all modalities simultaneously, such as emergency medical aid, multi-modal search engines, and financial data analysis. More commonly, data from all modalities is collected asynchronously. To improve the practical applicability of CMH, this paper proposes a novel method termed SHE to enhance the flexibility of processing asynchronously collected multimodal data.
>
> **R-Weakness-3**: To enhance the readability and understanding of our SHE method, we explain some loss functions in detail. 1) For the Knowledge Library Regularization loss (KLR), which is formulated as:
>
> $L_{klr} = -\frac{1}{CK} \sum_{c=1}^{C} \sum_{k=1}^{K} \log \frac{ \sum_{k'=1,k' \ne k }^{K}e^ {\Gamma(p^{c,k}_m,p^{c, k'}_m)}} {\sum _{c'=1}^{C}\sum _{k'=1}^{C}{e^ {\Gamma(p^{c,k}_m,p^{c',k'}_m)}}-e}.$
>
> KLR aims to escape a simple solution where all prototypes converge to a single point. As the above equation shows, we first treat inter-class prototypes as positive pairs and intra-class prototypes as negative pairs. Then, we maximize the similarity among positive pairs and minimize the similarity among negative pairs, thereby enhancing the distinctiveness between inter-class prototypes while ensuring semantic consistency between intra-class prototypes. 2) For the Knowledge Library Transfer loss (KLT), which is formulated as:
>
> $L_{klt} = -\frac{1}{CK} \sum_{c=1}^{C} \sum_{k=1}^{K}
> \log \left [ min(1,max(0,\Gamma(p_1^{c,k},p_m^{c,k}) - \sigma +1)) \right ].$
>
> KLT aims to transfer the semantic information in the newly mined knowledge to the benchmark knowledge library, thereby achieving cross-modal semantic consistency. As is shown in the above equation, we encourage the prototypes mined from the newly coming modality to align with the prototypes with the same semantics in the benchmark knowledge library by maximizing their similarity. Notably, we focus on maintaining a certain level of semantic similarity between them by setting a similarity boundary rather than forcing them to be identical. That is, the above learning objective is $\Gamma(p_1^{c,k},p_m^{c,k}) \ge \sigma$, which relaxes the alignment requirement and can further improve the representation ability of multimodal data.
>
> **R-Weakness-4**: Parallel training means that the proposed SHE can independently process each modality. When facing datasets with five modalities, SHE can directly perform training without any preprocessing, resulting in $5$ training processes and a storage overhead of $5$ sub-networks. By contrast, all baselines except MARS are limited to joint training on paired data from two modalities. When these methods face datasets with five modalities, we first construct pseudo-instance pairs through label-based repeat sampling and then conduct training on every two modalities. As a result, they require $5\times 4/2 = 10$ training processes and incur a storage overhead of $5 \times 4 = 20$ sub-networks. Obviously, parallel training can enhance training efficiency and reduce the required storage space. Besides, to further show the advantages of parallel training, we compare the training time (TT) and storage overhead (SO) of SHE and SCH on the XMedia dataset as follows:
>
> |Method|TT|SO|
> |:-:|:-:|:-:|
> | SCH |2148 s|2315.72 MB|
> | SHE |240 s|910.66 MB|
>
> The results show that our SHE can enhance training efficiency and reduce the required storage space by parallel training.
>
> **R-Suggestions**: We carefully check our paper and ensure that similar mistakes are not made.
>
> **Reference**
>
> [1] Liu X, Zeng H, Shi Y, et al. Deep cross-modal hashing based on semantic consistent ranking[J]. IEEE Transactions on Multimedia, 2023, 25: 9530-9542.
>
> [2] Luo H, Zhang Z, Nie L. Contrastive incomplete cross-modal hashing[J]. IEEE Transactions on Knowledge and Data Engineering, 2024, 36: 5823-5834.

---

> > ### Comment · Reviewer_TSiu · 2025-04-09
> >
> > Thank you for your detailed response. I have also read the comments from other reviewers. Most of my concerns have been adequately addressed.  As a result, I would like to keep my score as '3', leading to acceptance.

---

### Decision · Program_Chairs · 2025-05-01

**Decision:**

Accept (poster)

**Comment:**

This paper introduces SHE, a novel cross-modal hashing framework tailored for streaming-media scenarios where multimodal data arrive asynchronously. By integrating Knowledge Library Mining, Knowledge Library Transfer, and Discriminative Hash Learning, SHE enables parallel training across modalities while preserving semantic consistency. Extensive experiments on four benchmark datasets demonstrate its advantages in retrieval performance, scalability, and storage efficiency.

Reviewers agreed that the paper addresses a significant and previously underexplored problem with a well-motivated and technically sound solution. Strengths include its practical relevance, strong empirical results, and comprehensive ablation studies. Concerns regarding the clarity of loss function definitions and the handling of low-quality initial modalities were addressed in the rebuttal.